# DATA-EFFICIENT FINETUNING USING CROSS-TASK NEAREST NEIGHBORS

## ABSTRACT

Language models trained on massive prompted multitask datasets like T0 (Sanh et al., 2021) or FLAN (Wei et al., 2021a) can generalize to tasks unseen during training. We show that training on a carefully chosen subset of instances can outperform training on all available data on a variety of datasets. We assume access to a small number (250-1000) of unlabeled target task instances, select their nearest neighbors from a pool of multitask data, and use the retrieved data to train target task specific models. Our method is more data-efficient than training a single multitask model, while still outperforming it by large margins. We evaluate across a diverse set of tasks not in the multitask pool we retrieve from, including those used to evaluate T0 and additional complex tasks including legal and scientific document QA. We retrieve small subsets of P3 (the collection of prompted datasets from which T0's training data was sampled) and finetune T5 models that outperform the 3-billion parameter variant of T0 (T0-3B) by 3-30% on 12 out of 14 evaluation datasets while using at most 2% of the data used to train T0-3B. These models also provide a better initialization than T0-3B for few-shot finetuning on target-task data, as shown by a 2-23% relative improvement over few-shot finetuned T0-3B models on 8 datasets.

## 1 INTRODUCTION

Finetuning large models with data from a diverse set of tasks, augmented to include brief descriptions of the tasks (i.e., prompts) has been shown to help models generalize to unseen tasks (Wei et al., 2021a; Sanh et al., 2021). This cross-task generalization capability is particularly helpful in cases where it is expensive to collect labeled target task training sets. Prior work trained single models with as much prompted data as possible — for example, Sanh et al. (2021) train a model on roughly 11 million instances (counting different prompt variations). The training datasets were selected without using any information about the target tasks with the goal of allowing models to generalize to new tasks from instructions alone, making the evaluation "zero-shot". However, it is unclear if all the training data is required for doing well on any given target task. Furthermore, given that neural network models have previously been shown to suffer from negative interference (where in training on more datasets results in worse performance on certain downstream tasks) in multitask setups (Aribandi et al., 2022) and benefit from pretraining on domain-relevant data (Gururangan et al., 2020; Phang et al., 2018), it is possible that training only on relevant prompted data could further improve task generalization.

Based on this hypothesis, we seek to find small subsets of relevant training data in the massive pool of multitask data that cause the models to generalize better to a given target task than the rest of the pool. Manually finding relevant training data in a massive pool of data is infeasible since it is not obvious which of the source tasks are relevant for a given target task, and which instances are most relevant for target task generalization within a source task dataset (see Section 5.1). Hence we rely on a simple method to *automatically* select these subsets. Additionally, as only some samples within a given dataset may be relevant to a target task, we select per-instance rather than per-dataset, unlike prior work, which tries to identify useful datasets for transfer learning (Aribandi et al., 2022; Phang et al., 2018) and train on all data within the chosen datasets. We use a setup similar to contemporary work examining retrieval-augmented cross-task generalization (Lin et al., 2022): we assume access to a small number of *unlabeled* target task instances and use these to retrieve *cross-task nearest neighbors* - labeled instances from the massive pool of data most similar to our unlabeled target

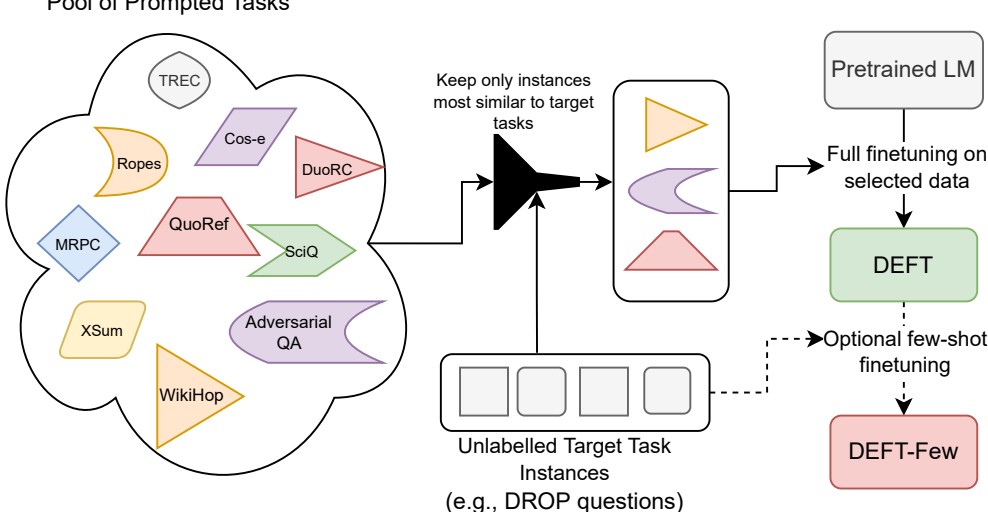

Figure 1: Overview of the DEFT method. Given some unlabeled target-task instances, we find the most similar instances in a large pool of multitask data. We train a model on these instances. If we have access to labeled data, we optionally few-shot finetune the DEFT model.

task instances. The similarity is computed as the distance between the representations produced by the encoder of a pretrained seq2seq model. Unlike prior work, we then finetune target task specific models on these neighbors alone, without using any target task specific labeled data or any extra data from the pool of multitask data. We hope that the similarity between the cross-task neighbors and our target task data will enable greater generalization to our target task, with dissimilar examples that may cause interference removed from the training mixture. We also aim to produce models that perform at least as well as models trained on the entire multitask pool despite being trained on a fraction of data, greatly reducing the cost of training.

We run experiments with T5 (Raffel et al., 2020) models, and use Public Pool of Prompts (P3) (Sanh et al., 2021) as the pool of prompted multitask data from which to retrieve cross-task nearest neighbors. We evaluate on the 11 datasets originally used to evaluate T0 (a collection of natural language understanding and commonsense tasks), as well as 3 additional datasets with varied domains (e.g., legal, NLP domains). Our findings are as follows:

- For all the target tasks, we find that their cross-task nearest neighbors in P3 are much more relevant as training data than the rest of the pool—training T5 models, sometimes even variants smaller than T0-3B, on these subsets yields models with performance 3-30% better than T0-3B evaluated zero-shot across 12 out of 14 datasets.

- For some target tasks on which T0-3B performs close to random chance, T5 models of the same size trained using cross-task nearest neighbors perform significantly above chance, confirming our hypothesis that massive multitask prompted training could lead to negative interference between tasks.

- When target task labeled data is available for few-shot finetuning, we find that T5 models trained with cross-task nearest neighbors provide better initialization for parameter-efficient finetuning methods than T0-3B, performing 2-23% better than T0-3B with few-shot finetuning across 10 out of 11 datasets.

- An analysis of what relevant data gets retrieved shows that most of the tasks in the massive pool of multitask data are not retrieved for any target tasks, confirming our hypothesis that only a small subset of data within the pool is relevant to any given target task.

These findings suggest that training these models on all the available multitask prompted data results in negative interference, even in relatively large (3 billion parameter) models. Furthermore,

they show that training large models on all available multitask data is not optimal, with higher performance possible through better-curated and much smaller training sets.

## 2 RELATED WORK

**Multi-task Transfer Models**   Training on large multi-task mixtures is a common trend within NLP, with most existing approaches first training a pretrained language model on a large collection of tasks, and then evaluating these models in either zero or few-shot settings on a collection of held-out datasets (Wei et al., 2021a; Sanh et al., 2021; Khashabi et al., 2020; Mishra et al., 2021; Aribandi et al., 2022). Most approaches do not customise their task selection to downstream tasks and assume no knowledge of the target tasks ahead of time, instead focusing on building a single model most applicable to any arbitrary evaluation task. In contrast, we show that if assume access to unlabeled target task instances, we can make much better use of the multitask data, selecting only instances useful to a given task. Relatedly, Vu et al. (2020) propose a method for using gradients from labelled task data to construct task embeddings for predicting task transferability. Our method instead uses unlabelled data, which is much cheaper and easier to collect, and does not use gradients, making it much easier to scale to large models such as T5-XL.

**Retrieval-based methods for NLP**   Adding retrieval components to language models has been shown (Khandelwal et al., 2019; Guu et al., 2020; Lewis et al., 2020) to augment their generalization capabilities by externalizing memorization. In contrast to prior work in this direction that mostly focused on language modeling as the end task, we evaluate on a variety of language understanding tasks. The work from Shi et al. (2022) used retrieval-based methods for classification tasks by heuristically mapping the label space of the end-tasks to that of the predicted next words of the nearest neighbors from a language model. We instead finetune the models on the nearest neighbors. Lin et al. (2022) also use unlabelled examples to retrieve relevant data for improving performance, but focus on *further augmenting multi-task models*, and limit themselves to small query sets and small retrieval sets. The authors claim that using a multi-task model is crucial for good retrieval performance. In contrast, we assume *no access to multi-task models*, showing that using non-multitask-trained models still work well for retrieval, and focus on *pruning the amount of multi-task data used during training*. Additionally, we use larger query sets (up to 1000 unlabelled examples) and retrieve thousands of examples (as opposed to hundreds).

**Parameter-efficient fine-tuning methods**   In contrast to work that focused on finetuning fewer parameters in large models to adapt them to new tasks (Houlsby et al., 2019; Hu et al., 2021; Liu et al., 2022), our proposal is a *data*-efficient pretraining method for obtaining task-specific models without using any target task labels. Our method is complementary to parameter-efficient methods, and they can both be used in conjunction, as shown in section 4.3.

**Instance attribution**   Our approach works by identifying the most relevant training examples for a given data point, which is called *instance attribution*. Prior work (Koh & Liang, 2017; Yeh et al., 2018; Pruthi et al., 2020; Han & Tsvetkov, 2022) used instance attribution methods to interpret predictions of neural network models. These methods generally relied on the gradients of the model to identify the effect specific data points, either in pretraining or finetuning stage, have on the model's predictions. Our method for identifying cross-task neighbors is simpler because we do not use gradients and we do not even rely on the labels of the data. Results from Pezeshkpour et al. (2021) show that instance attribution based on similarity between the model's representations are comparable to gradient-based approaches in terms of finding the most important training data points.

## 3 DATA EFFICIENT FINETUNING ACROSS MULTIPLE TASKS

Given a large collection of labeled prompted data (i.e., data converted into a text-to-text form, with task instructions included in the plain-text input, e.g., P3), our core hypothesis is that some tasks in this massive pool of data are more similar to a given target task than others. Given a target task, we assume we have access to a small amount of *unlabeled* target task data, which is often much easier and cheaper to collect than labeled data (see section 5.2). Our aim is to find a relevant subset of data

from our pool given a single target task, ideally allowing us to train a model using this subset that outperforms a similar model trained on the entire pool of data.

Manually identifying the relevant subsets of these datasets is not feasible since task boundaries are usually not clearly defined in NLP, and it is hard to interpret what skills get transferred when a model is trained on one dataset and evaluated on other. Hence, we use similarity between the pretrained model's representations to compute relevance. We encode all instances in the large pool of multitask data with a pretrained language model (e.g., T5), and build a search index over the resulting representations. Given small amounts of unlabeled target task data, we retrieve relevant multitask subsets from the index, which we call **cross-task nearest neighbors** of the target tasks. We then build task-specific models by finetuning the pretrained models on the cross-task neighbors. We refer to this approach as **Data-Efficient FineTuning**, or **DEFT** for short.

We evaluate our approach both in cases where no labeled data is available, and when a few (20-70) annotated labels are available. In the former case, we simply use the unlabeled data for retrieval and evaluate the resulting DEFT model 'zero-shot' on the target task. In the latter case, we first train a DEFT model and then perform parameter-efficient few-shot tuning using IA3 (Liu et al., 2022) to make use of the labeled data.

**Retrieving cross-task nearest neighbors**    To retrieve the most similar instances to a given set of target task instances, we first build an index over the massive pool of multi-task data for efficient retrieval, encoding samples using a pretrained encoder. Then, given a set of query instances $Q$, we retrieve our subset of similar data by computing a union of the $k$-nearest neighbors to all $q \in Q$. Note that there may be an overlap between the sets of nearest neighbors retrieved for different queries, and hence $|R| \leq |Q| * k$, where $R$ is the retrieved subset. Empirically, we find $|R|$ tends to be 5-50× smaller than $|Q| * k$ due to this overlap.

**Data-Efficient Training**    Given a retrieved set of data $R$, we can then simply train a pretrained language model on this mixture of data using a cross-entropy loss, as all data is in a unified text-to-text prompted format. This training is similar to the multitask prompted training of T0 (Sanh et al., 2021). We refer to models trained on $R$ as DEFT models. If we have no labeled data available, we directly evaluate these models on our target tasks.

**Parameter-efficient Few-shot Finetuning**    For the case where few annotated labels are available, we make use of parameter-efficient few-shot finetuning. For this, we take our multi-task trained DEFT checkpoints and finetune them using IA3 (Liu et al., 2022) on task-specific few-shot data. Concretely, given a trained transformer model, we introduce three new vectors $l_k$, $l_v$, and $l_{ff}$ into the attention and feed-forward mechanisms of each layer as follows:

$$\text{Attention}(Q, K, V) = \text{softmax}\left(\frac{Q(l_k \odot K^T)}{\sqrt{d_k}}\right)(l_v \odot V) \tag{1}$$

$$\text{Feed-forward}(x) = (l_{ff} \odot f(W_1 x))W_2 \tag{2}$$

We initialize these vectors with all ones and only update them during the few-shot finetuning. This provides an efficient method of further training our DEFT models on task-specific data and has been shown to outperform full finetuning in the few-shot setting (Liu et al., 2022).

## 4 EXPERIMENTS

### 4.1 SETUP & HYPERPARAMETERS

**Indexing P3**    We construct an index of P3 data using FAISS (Johnson et al., 2019), a library for efficient similarity search over dense vectors. We use a Hierarchical Navigable Small World index (Malkov & Yashunin, 2020) to approximate the $k$-nearest neighbor search. To keep the size of the index manageable, we use Product Quantization Jegou et al. (2010) and reduce the dimensionality of the encoded representations using an Optimized Product Quantization transform Ge et al. (2013). We encode our instances using the T5 v1.1 model with extra language model pretraining introduced by Lester et al. (2021). For all experiments, we match the size of the encoder used to index data and the size of downstream models trained on this data (e.g., if we train a T5-XL sized

model, we use T5-XL to index and retrieve the data). We use the subset of P3 used to train T0 as our pool of multitask data unless otherwise stated.

**DEFT Training**  Following T0, we start with the T5 v1.1 model with extra language model pre-training. Unless otherwise stated, we use the 'XL' variant with 3 billion parameters across our experiments. When training on cross-task nearest neighbors, we train for 5 epochs with a batch size of 8 using the Adam optimizer (Kingma & Ba, 2015) and a learning rate of 5e-5. We use a linear warmup schedule for the first 10% of the total training steps and linear decay for the rest of training.

**Few-shot training**  We follow the settings suggested by Liu et al. (2022): training for 1000 steps with a batch size of 8. We use the Adafactor optimizer with a maximum learning rate of 3e-3 and a linear decay schedule with 60 warmup steps. We only update the IA3 vectors during training.

**Evaluation Datasets**  We evaluate on the set of 11 datasets used to evaluate T0 (RTE, ANLI R1/2/3, CB, HellaSwag, Story Cloze, WinoGrande, WSC, COPA, WiC), which include natural language inference and commonsense reasoning datasets. In addition to the T0 evaluation datasets, we also evaluate on three additional datasets from diverse domains: CaseHold (Chalkidis et al., 2022; Zheng et al., 2021), a legal QA dataset, DROP (Dua et al., 2019), a QA dataset that requires discrete operations, and a subtask of Qasper (Dasigi et al., 2021), a QA dataset over entire NLP papers. Qasper has two subtasks—selecting paragraphs in the paper that provide evidence for answering the questions, and generating the answers. We focus on the former because it was shown to more difficult of the two, and convert it into a binary classification task where the inputs are combinations of questions and single paragraphs. We refer to this subtask as *QasperEvidence* henceforth, and evaluate model performance in terms of document-level F1 as described in Dasigi et al. (2021). For evaluation and few-shot training, we convert all datasets to a prompted text-to-text format[1] either using the prompt templates from P3 for the T0 evaluation datasets or an original prompt for the other datasets. For CaseHold, DROP, and QasperEvidence we randomly split out 1000 examples from the existing validation sets to use for retrieval, and use the remaining data for evaluation. For all other datasets, we simply retrieve using up to 1000 randomly chosen examples from the training splits (if a dataset has less than 1000 training examples, we use all available training data for retrieval). We provide a list of these prompts and further dataset details in Appendix A.

**Model Evaluation**  Following Sanh et al. (2021) and Brown et al. (2020), we calculate accuracy on all datasets except DROP using *rank classification*, where we pick the answer with lowest loss across possible answer choices given the instance input as the model prediction. As DROP is an QA dataset that requires selecting spans or generating numbers, and does not have answer choices, we simply generate the prediction using greedy decoding.

**Baselines**  For zero-shot evaluation, we primarily compare against 4 baselines: 1) *T0-3B*, trained on about 10% of the P3 data[2], 2) *Random*, a model trained on a random selection of P3 data the same size as the subsets selected by DEFT, 3) *T5-XL* not finetuned any further, and 4) *BM25*, using BM25[3] (Robertson & Zaragoza, 2009) for retrieval instead of dense representations. For few-shot settings, we compare T0-3B with additional few-shot training with DEFT checkpoints trained on subsets chosen using (a) 1000 unlabeled instances and (b) the instances used in the few-shot training without labels. This means (b) uses no additional data, labeled or unlabeled, compared to T0-3B with few-shot finetuning.

## 4.2 DATA-EFFICIENT FINE-TUNING VS. MASSIVE MULTITASK TRAINING

We first assume we have access *only* to unlabeled task-specific data, and so cannot train on any target task labeled data. We sample 1000 unlabeled instances per dataset and retrieve the 500 nearest

---

[1]For example, ANLI instances were converted to '{premise} Question: {hypothesis} True, False, or Neither?', with the answers as 'true', 'false', or 'neither'.

[2]Sanh et al. (2021) report that they train T5-XL on at most 500K instances per prompted dataset in P3, which amounts to about 10% of the pool.

[3]We use Pyserini (Lin et al., 2021) with default settings for building and retrieving from the BM25 index. We restrict the amount of data retrieved to same size as the subsets retrieved by DEFT.

| Task | DEFT-XL | T0-3B | Rand-XL | Rand-Bal | T5-XL | BM25-XL | ReCross* | DEFT-base | Rand-base | T5-base | Maj. Class |
|---|---|---|---|---|---|---|---|---|---|---|---|
| CaseHold | 37.2 | 30.9 | 19.0 | **38.7** | 11.4 | 27.9 | - | **18.9** | 17.5 | 11.4 | 6.6 |
| DROP | **31.0** | 27.4 | 24.3 | 27.6 | 11.3 | 22.6 | - | **21.3** | 18.0 | 4.0 | - |
| QasperEv. | **28.5** | 19.9 | 17.9 | 23.2 | 8.2 | 20.3 | - | **15.9** | 11.0 | 8.2 | 19.9 |
| RTE | 74.0 | 70.4 | **78.3** | 78.0 | 53.1 | 74.3 | - | **61.7** | 61.0 | 52.0 | 53.4 |
| ANLI R1 | 39.8 | 35.0 | 35.3 | **40.0** | 32.9 | 37.5 | - | 29.6 | **33.3** | 32.9 | 33.4 |
| ANLI R2 | **37.5** | 32.6 | 35.3 | 36.9 | 33.5 | 36.9 | - | 32.5 | 22.3 | **33.5** | 33.4 |
| ANLI R3 | 41.4 | 35.3 | 38.0 | **41.7** | 33.8 | 41.1 | 30.2 | 31.6 | **33.1** | 32.7 | 33.5 |
| CB | **60.7** | 58.9 | **60.7** | 55.4 | 44.6 | 50.0 | 31.6 | **50.0** | 48.2 | 44.6 | 50.0 |
| HellaSwag | **33.1** | 28.2 | 27.4 | 29.3 | 23.0 | 28.7 | 26.0 | **25.9** | 25.0 | 23.0 | 25.7 |
| StoryCloze | **95.3** | 86.5 | 79.1 | 94.1 | 53.0 | 82.3 | - | **83.5** | 57.4 | 53.0 | 51.4 |
| WinoGrande | 50.6 | 50.0 | 49.2 | 49.2 | **50.8** | 50.1 | 54.4 | **50.8** | 50.1 | **50.8** | 50.4 |
| WSC | 39.4 | **50.0** | 47.1 | 46.2 | 36.3 | 36.5 | 61.4 | **42.3** | 36.5 | 36.3 | 63.5 |
| COPA | **95.0** | 74.0 | 80.0 | 88.0 | 60.0 | 79.0 | - | **66.0** | 44.0 | 60.0 | 55.0 |
| WiC | 54.9 | 51.1 | 51.4 | **57.5** | 51.7 | 51.9 | 50.6 | 49.4 | 50.0 | **51.7** | 50.0 |
| Average | **51.3** | 46.5 | 45.9 | 50.4 | 35.9 | 45.7 | - | **41.4** | 37.0 | 35.3 | - |

Table 1: Performance of XL (3B) and base size (∼250 million) models across datasets. 'Rand' refers to performance of models trained on randomly chosen P3 subsets of equivalent size to the ones chosen by DEFT, with 'Rand-bal' using uniform random sampling across tasks for subset selection. 'T5' refers to performance of a non-finetuned T5 model. 'BM25' refers to models trained on subsets of equivalent size to DEFT subsets from P3 retrieved using BM25. DROP and QasperEv. Results are F1 scores, CaseHold micro F1, all else accuracy. Datasets below dashed line were used to evaluate T0 in Sanh et al. (2021). * ReCross results reported by Lin et al. (2022) and are mean performance across multiple prompts.

neighbors[4] of each instance. We then train dataset-specific models on each of the retrieved sets. As seen in Table 1, our DEFT-XL models consistently outperform T0-3B and other baselines, with a median relative improvement of 13% over T0-3B. We also see that T5-base sized models also improve over baselines in Table 1 - the DEFT-base models have a median relative improvement of 8% over the random baseline. All DEFT models are trained on subsets of P3 consisting of 0.1-2% of all P3 data. This confirms our hypothesis that training on a well-chosen subset of P3 is more beneficial for target task performance than training on a uniform sample of all available data. We also note that using dense representations is crucial to our approach, as using BM25 for retrieval instead overall underperforms T0-3B and even the random baselines. We also note DEFT-XL outperforms ReCross by large margins for 4/6 tasks. Our results suggest that a general language model encoder can still retrieve relevant cross-task neighbors, contrary to the claims made by Lin et al. (2022).

Remarkably, DEFT-XL outperforms the majority baselines on two target datasets (QasperEvidence and ANLI R2) where T0-3B does not, and DEFT-base on one (COPA). This observation further confirms that multitask models trained on uniformly sampled data might be suffering from negative interference between tasks.

We note our approach underperforms compared to T0-3B and Rand-XL for WSC and RTE respectively. However, both datasets have small evaluation sets and large variance (see Appendix B), leading us to believe these differences are not significant. Relatedly, T0-3B underperforms a majority class baseline on WSC.

### 4.3 FEW-SHOT FINETUNING OF DEFT MODELS

Next, we assume we are able to label a small number of task-specific examples, and further train our DEFT models. We reuse the XL-size models trained in section 4.2 and further train them using the parameter-efficient IA3 on the few-shot data used by Liu et al. (2022). As seen in table 2, DEFT models with few-shot finetuning ('DEFT-Few (1kQ)') perform on average 7% better than T0-3B models with few-shot finetuning ('T0-3B+IA3'), with statistically significant gains on 5 datasets. This shows that DEFT models serve as better starting points for few-shot finetuning than T0-3B, providing similar or better performance across all datasets despite being exposed to much less training data. Notably, DEFT-Few significantly outperforms T0-3B+IA3 on WinoGrande, for which zero-shot DEFT did not significantly outperform zero-shot T0-3B. These results show that DEFT models are more amenable to few-shot fine-tuning than T0-3B.

---

[4]We retrieve 2500 nearest neighbours for T5-base, as early experiments suggested it performed worse with only 500 neighbors.

| | T0-3B+IA3 | T5+IA3 | Rand+IA3 | DEFT-Few (1kQ) | DEFT-Few (20-70Q) |
|---|---|---|---|---|---|
| RTE | $77.5_{2.0}$ | $57.0_{4.3}$ | $83.3_{1.1}$ | $79.4_{1.3}$ | $81.3_{1.6}$ |
| ANLI R1 | $44.9_{3.0}$ | $39.6_{1.8}$ | $43.3_{2.3}$ | $47.3_{1.4}$ | $47.3_{1.5}$ |
| ANLI R2 | $39.5_{1.7}$ | $36.5_{1.4}$ | $40.3_{1.6}$ | $40.8_{2.8}$ | $42.2_{2.7}$ |
| ANLI R3 | $40.2_{2.2}$ | $34.8_{1.1}$ | $39.3_{2.3}$ | $44.3_{2.1}$ | $42.9_{1.8}$ |
| CB | $78.9_{3.9}$ | $67.9_{2.5}$ | $81.4_{3.5}$ | $82.5_{2.6}$ | $84.6_{4.3}$ |
| HellaSwag | $34.7_{0.6}$ | $27.5_{1.1}$ | $38.1_{1.1}$ | $42.5_{2.1}$ | $45.9_{1.8}$ |
| StoryCloze | $93.0_{0.6}$ | $83.0_{3.1}$ | $92.6_{0.8}$ | $96.2_{0.2}$ | $96.5_{0.2}$ |
| WinoGrande | $50.6_{1.3}$ | $49.8_{0.8}$ | $51.4_{2.3}$ | $55.9_{3.0}$ | $55.2_{3.1}$ |
| WSC | $64.8_{3.5}$ | $51.0_{1.0}$ | $55.8_{3.0}$ | $63.3_{5.2}$ | $59.6_{3.8}$ |
| COPA | $82.0_{2.7}$ | $61.6_{4.2}$ | $86.6_{1.7}$ | $95.4_{1.5}$ | $92.6_{2.2}$ |
| WiC | $54.9_{1.9}$ | $56.6_{3.0}$ | $54.5_{2.4}$ | $57.7_{2.9}$ | $57.4_{2.9}$ |
| Average | 60.1 | 51.4 | 60.6 | **64.1** | **64.1** |

Table 2: Performance of IA3 few-shot finetuned models using XL-size checkpoints. For all models we report the mean over 5 runs with the standard deviation as subscript. We report performance for DEFT-Few models using 1000 unlabeled queries ('1kQ') and few-shot queries ('20-70Q'). See section 4.3 for details. **Bolded** numbers indicate that value is statistically significantly better than T0-3B+IA3 ($p < .05$ in a two-tailed paired t-test).

**Few-shot Retrieval** In this experiment, we evaluate DEFT in a setting where we have access only to a small number of target-task labeled examples (exactly what is available to T0-3B+IA3), and no additional unlabeled examples. We construct 5 few-shot sets for each dataset, and for each set retrieve cross-task neighbors using the few-shot data, finetune T5 models on the retrieved data, and then finally finetune using IA3 on the labeled few-shot data itself. In order to make up for the smaller query set, we retrieve the closest 2000 neighbors per query instance. As seen in Table 2, this still results in a model that outperforms T0-3B with few-shot tuning ('DEFT-Few (20-70Q)'), and overall achieves similar performance to DEFT-Few (1kQ). Crucially, this shows that DEFT followed by few-shot finetuning may be a better alternative to few-shot finetuning T0-3B even when both methods have *exactly* the same target-task information available.

## 5 ANALYSIS

### 5.1 CROSS-TASK RETRIEVAL

**What gets retrieved?** We show what source datasets get selected during retrieval for each evaluation dataset in figure 2. For most target datasets, the majority of source datasets are *not* selected, further strengthening our hypothesis that much of the massive multitask pool is not relevant to a given target task, and no single mixture of datasets is optimal for all target tasks. We additionally find that no more than 27% of all instances within any source dataset is retrieved, suggesting that our approach is also effective at finding relevant subsets of data *within* large datasets.

**Retrieval Hyperparameters** When retrieving cross-task data, the amount and quality of data retrieved is highly dependent on the *query size* (i.e. the number of task-specific instances used for retrieval) and *number of neighbors* (i.e. the number of cross-task samples retrieved per task-specific instance). In figure 3, we show the effect of varying both query size (sweeping from 32 to all training data) and the number of neighbors (sweeping from 1 to 5000) on dataset performance on RTE and CaseHold respectively. We find that increasing the amount of data retrieved, whether through increasing the number of neighbors or query set size, results in improved performance up to a point, and then either plateaus or decreases, providing evidence for our hypothesis that using 'too much' data can result in reduced downstream performance due to negative interference.

**What model should you use for retrieval?** To determine the effect of of model size on indexing and retrieval, we train models using the cross-task neighbors retrieved by base and XL-size models when the query size and number of neighbors is held constant. We find that using a larger (XL size) indexing model generally results in better performance, but this gap is much larger when training a base size model (8%) than when training XL-size models (1%), suggesting that smaller models benefit more from larger retrieval models. We provide detailed results in Appendix C.

| | anli_r1 | anli_r2 | anli_r3 | casehold | cb | copa | drop | hellaswag | qasper | rte | story_cloze | wic | winogrande | wsc | P3 |
|---|---|---|---|---|---|---|---|---|---|---|---|---|---|---|---|
| adversarial_qa | 0.28 | 0.28 | 0.11 | 0.15 | 0.04 | 0.00 | 0.47 | 0.07 | 0.24 | 0.09 | 0.00 | 0.01 | 0.00 | 0.03 | 0.02 |
| ag_news | 0.02 | 0.03 | 0.15 | 0.00 | 0.03 | 0.00 | 0.01 | 0.00 | 0.00 | 0.27 | 0.00 | 0.00 | 0.00 | 0.00 | 0.04 |
| amazon_polarity | 0.04 | 0.03 | 0.03 | 0.01 | 0.02 | 0.00 | 0.00 | 0.08 | 0.04 | 0.02 | 0.00 | 0.00 | 0.00 | 0.00 | 0.04 |
| app_reviews | 0.00 | 0.00 | 0.00 | 0.00 | 0.00 | 0.00 | 0.00 | 0.00 | 0.00 | 0.00 | 0.00 | 0.00 | 0.00 | 0.00 | 0.04 |
| cnn_dailymail_3.0.0 | 0.01 | 0.00 | 0.08 | 0.27 | 0.02 | 0.00 | 0.05 | 0.05 | 0.01 | 0.08 | 0.01 | 0.00 | 0.00 | 0.00 | 0.04 |
| common_gen | 0.00 | 0.00 | 0.00 | 0.00 | 0.00 | 0.00 | 0.00 | 0.00 | 0.00 | 0.00 | 0.00 | 0.03 | 0.05 | 0.01 | 0.04 |
| cos_e_v1.11 | 0.00 | 0.00 | 0.00 | 0.00 | 0.02 | 0.53 | 0.00 | 0.01 | 0.01 | 0.00 | 0.00 | 0.01 | 0.02 | 0.01 | 0.01 |
| cosmos_qa | 0.00 | 0.00 | 0.07 | 0.00 | 0.26 | 0.02 | 0.00 | 0.22 | 0.01 | 0.00 | 0.32 | 0.00 | 0.02 | 0.18 | 0.04 |
| dbpedia_14 | 0.09 | 0.10 | 0.01 | 0.00 | 0.00 | 0.00 | 0.00 | 0.00 | 0.00 | 0.00 | 0.00 | 0.00 | 0.00 | 0.00 | 0.04 |
| dream | 0.00 | 0.00 | 0.00 | 0.00 | 0.16 | 0.02 | 0.00 | 0.01 | 0.01 | 0.00 | 0.01 | 0.01 | 0.02 | 0.03 | 0.00 |
| duorc_ParaphraseRC | 0.01 | 0.01 | 0.00 | 0.01 | 0.00 | 0.00 | 0.00 | 0.01 | 0.00 | 0.00 | 0.00 | 0.00 | 0.00 | 0.01 | 0.04 |
| duorc_SelfRC | 0.01 | 0.01 | 0.00 | 0.04 | 0.00 | 0.00 | 0.00 | 0.01 | 0.00 | 0.00 | 0.00 | 0.00 | 0.00 | 0.00 | 0.04 |
| gigaword | 0.00 | 0.00 | 0.00 | 0.00 | 0.00 | 0.00 | 0.00 | 0.00 | 0.00 | 0.01 | 0.00 | 0.00 | 0.00 | 0.00 | 0.04 |
| glue_mrpc | 0.00 | 0.00 | 0.01 | 0.00 | 0.00 | 0.00 | 0.00 | 0.00 | 0.00 | 0.03 | 0.00 | 0.05 | 0.02 | 0.02 | 0.00 |
| glue_qqp | 0.00 | 0.00 | 0.01 | 0.00 | 0.01 | 0.01 | 0.00 | 0.01 | 0.21 | 0.03 | 0.00 | 0.60 | 0.00 | 0.01 | 0.04 |
| imdb | 0.00 | 0.00 | 0.00 | 0.00 | 0.00 | 0.00 | 0.00 | 0.00 | 0.00 | 0.00 | 0.00 | 0.00 | 0.00 | 0.00 | 0.04 |
| kilt_tasks_hotpotqa | 0.18 | 0.19 | 0.06 | 0.00 | 0.00 | 0.00 | 0.00 | 0.00 | 0.00 | 0.01 | 0.00 | 0.00 | 0.00 | 0.00 | 0.04 |
| multi_news | 0.00 | 0.00 | 0.02 | 0.30 | 0.01 | 0.00 | 0.02 | 0.02 | 0.02 | 0.03 | 0.00 | 0.00 | 0.00 | 0.00 | 0.03 |
| paws_labeled_final | 0.08 | 0.06 | 0.03 | 0.00 | 0.02 | 0.00 | 0.00 | 0.00 | 0.05 | 0.09 | 0.00 | 0.11 | 0.09 | 0.13 | 0.04 |
| qasc | 0.00 | 0.00 | 0.01 | 0.00 | 0.03 | 0.02 | 0.00 | 0.00 | 0.03 | 0.02 | 0.00 | 0.06 | 0.06 | 0.05 | 0.01 |
| quail | 0.00 | 0.00 | 0.00 | 0.08 | 0.03 | 0.00 | 0.00 | 0.12 | 0.03 | 0.00 | 0.00 | 0.00 | 0.00 | 0.00 | 0.02 |
| quarel | 0.00 | 0.00 | 0.02 | 0.00 | 0.01 | 0.01 | 0.00 | 0.00 | 0.03 | 0.00 | 0.02 | 0.03 | 0.05 | 0.10 | 0.00 |
| quartz | 0.00 | 0.00 | 0.01 | 0.00 | 0.00 | 0.00 | 0.00 | 0.00 | 0.01 | 0.01 | 0.00 | 0.02 | 0.02 | 0.02 | 0.00 |
| quoref | 0.01 | 0.01 | 0.00 | 0.03 | 0.00 | 0.00 | 0.21 | 0.01 | 0.01 | 0.00 | 0.00 | 0.00 | 0.00 | 0.00 | 0.02 |
| ropes | 0.02 | 0.02 | 0.04 | 0.01 | 0.02 | 0.00 | 0.04 | 0.27 | 0.08 | 0.01 | 0.07 | 0.00 | 0.01 | 0.03 | 0.01 |
| rotten_tomatoes | 0.00 | 0.00 | 0.00 | 0.00 | 0.00 | 0.00 | 0.00 | 0.00 | 0.00 | 0.00 | 0.00 | 0.00 | 0.00 | 0.00 | 0.01 |
| samsum | 0.00 | 0.00 | 0.01 | 0.00 | 0.12 | 0.00 | 0.00 | 0.00 | 0.00 | 0.00 | 0.05 | 0.00 | 0.00 | 0.00 | 0.01 |
| sciq | 0.01 | 0.00 | 0.01 | 0.00 | 0.00 | 0.01 | 0.00 | 0.02 | 0.02 | 0.00 | 0.00 | 0.01 | 0.00 | 0.01 | 0.01 |
| social_i_qa | 0.00 | 0.00 | 0.05 | 0.00 | 0.17 | 0.39 | 0.00 | 0.05 | 0.01 | 0.01 | 0.52 | 0.03 | 0.62 | 0.32 | 0.02 |
| trec | 0.00 | 0.00 | 0.00 | 0.00 | 0.00 | 0.00 | 0.00 | 0.00 | 0.03 | 0.00 | 0.00 | 0.01 | 0.00 | 0.00 | 0.00 |
| wiki_bio | 0.00 | 0.00 | 0.00 | 0.00 | 0.00 | 0.00 | 0.00 | 0.00 | 0.00 | 0.00 | 0.00 | 0.00 | 0.00 | 0.00 | 0.04 |
| wiki_hop_original | 0.04 | 0.04 | 0.01 | 0.01 | 0.00 | 0.00 | 0.11 | 0.00 | 0.01 | 0.00 | 0.00 | 0.00 | 0.00 | 0.00 | 0.04 |
| wiki_qa | 0.17 | 0.18 | 0.11 | 0.01 | 0.02 | 0.00 | 0.00 | 0.00 | 0.15 | 0.13 | 0.00 | 0.01 | 0.00 | 0.01 | 0.01 |
| wiqa | 0.00 | 0.00 | 0.00 | 0.00 | 0.00 | 0.00 | 0.00 | 0.01 | 0.00 | 0.00 | 0.00 | 0.00 | 0.00 | 0.00 | 0.03 |
| xsum | 0.01 | 0.01 | 0.13 | 0.07 | 0.01 | 0.00 | 0.07 | 0.01 | 0.01 | 0.15 | 0.00 | 0.00 | 0.00 | 0.00 | 0.04 |
| yelp_review_full | 0.00 | 0.00 | 0.00 | 0.00 | 0.00 | 0.00 | 0.00 | 0.01 | 0.00 | 0.00 | 0.00 | 0.00 | 0.00 | 0.00 | 0.04 |

Figure 2: Proportion of the retrieved training data for each evaluation dataset (columns) that comes from each dataset in P3 (rows). The final column shows these values for all of P3.

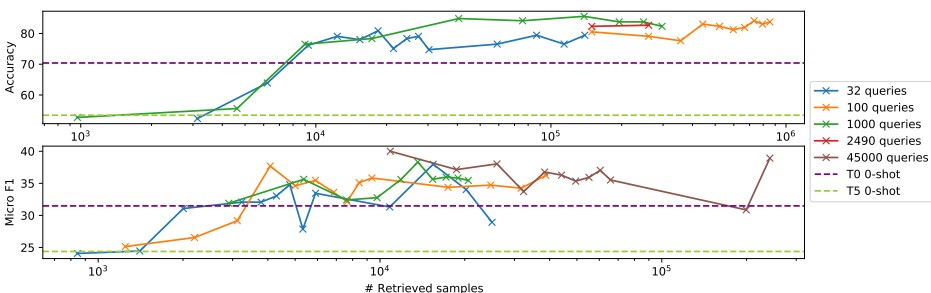

Figure 3: (above) RTE accuracy by number of P3 samples retrieved for DEFT-XL across a varying number of neighbors. (below) CaseHold micro F1 by number of P3 samples retrieved for DEFT-XL across a varying number of neighbors.

**Are prompts useful for retrieval?** All P3 data is in a prompted format, where the input is made up of (a) the input instance and (b) a prompt that contains information about the task. Training on prompted data greatly aids zero-shot generalisation (Wei et al., 2021b; Sanh et al., 2021), but it is unclear how useful it is for retrieval. To examine this, we run experiments using Super-Natural

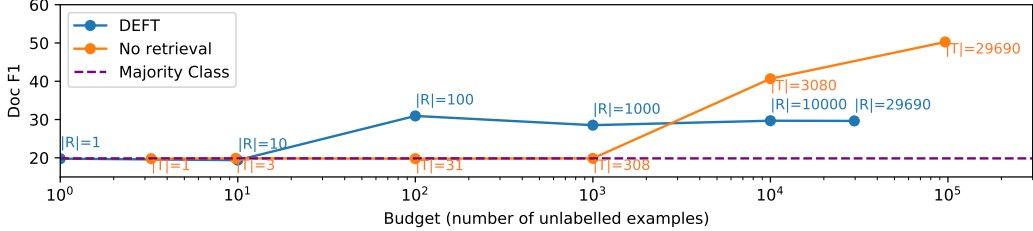

Figure 4: Performance of DEFT-XL and full-finetuning methods with the same annotation budget used for obtaining either labeled or unlabeled data for QasperEvidence. Unless one has a large annotation budget, collecting unlabelled examples is superior to collecting labelled ones. $|R|$ and $|T|$ refer to the size of the retrieval and train sets respectively. See section 5.2 for details.

Instructions (Wang et al., 2022), a recent instruction-tuning dataset. We index and retrieve the data with and without instructions in the input and compare the performance after training on retrieved subsets[5]. We find that retrieving **without** instructions outperforms retrieving with instructions by a small margin (49.2 vs 49.5 average performance), suggesting that DEFT relies more on instance information rather than task information for retrieval. We provide more details in Appendix D.

## 5.2 PRACTICALITY OF ASSUMING ACCESS TO UNLABELED DATA

Contrary to prior work, our approach assumes access to unlabeled data. This is a practical assumption given that unlabeled data is often readily available or is far cheaper to acquire than labeled data. This is especially true for tasks such as Qasper or CaseHold, which require experts to carefully read (sometimes quite long) texts to provide labels. We argue that DEFT's use of unlabeled data can make it a cost-efficient method to obtain a well-performing task-specific model when the data labeling budget is limited.

We examine this by studying a scenario where QasperEvidence data was collected and assume we have access to P3 and DEFT to make efficient use of it. Obtaining labeled instances for QasperEvidence cost 3.25 times acquiring unlabeled (question-paragraph) instances[6]. We compare (Figure 4) performance on the test set of a T5-XL model trained on a varying number of labeled instances with a DEFT-XL model trained on cross-task nearest neighbors of 3.25 as many unlabeled instances. DEFT yields better results for smaller annotation budgets ($< 1000$ labelled examples), and underperforms models trained on a few thousand labelled examples. This confirms our suggestion that DEFT is preferable to regular finetuning for limited budgets. We also note that the DEFT setup makes it easy to use target-task labeled data when available, as shown in section 4.3.

## 6 CONCLUSION

In this work, we propose Data-Efficient FineTuning, a novel method for efficiently using multitask data by training task-specific models using only a small amount of unlabeled target task data. We use the unlabeled data to select subsets of the multitask data, and train models on these subsets. The resulting models outperform same-sized models trained on all available data (e.g., T0), suggesting that negative interference is still present even in large massively multi-task models. Our approach also results in models that work as better starting points for few-shot finetuning, outperforming models trained on more data. Our approach still performs strongly even when as few as only 20 unlabeled examples are available. Overall, our results strongly suggest that training on all available data, even with large models, is not the optimal choice for most downstream tasks, and that instead focusing on ways to better curate datasets into higher-quality, but smaller, mixtures is potentially a better path forward.

---

[5]We add instructions back into samples without them in order to isolate the effect of instructions on retrieval separate from their effect during finetuning.

[6]Based on an estimate provided by the authors of the dataset. Questions were written after only the abstracts of papers were read, and evidence selection required reading entire papers.

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

| Dataset | Retrieval | Eval | #Shots | Retrieval from |
|---|---|---|---|---|
| CaseHold (Zheng et al., 2021) | 1000 | 2900 | - | Validation |
| DROP (Dua et al., 2019) | 1000 | 8535 | - | Validation |
| QasperEvidence (Dasigi et al., 2021) | 1000 | 43673 | - | Validation |
| RTE* | 1000 | 277 | 32 | Train |
| ANLI R1 (Nie et al., 2020) | 1000 | 1000 | 50 | Train |
| ANLI R2 (Nie et al., 2020) | 1000 | 1000 | 50 | Train |
| ANLI R3 (Nie et al., 2020) | 1000 | 1000 | 50 | Train |
| CB (De Marneffe et al., 2019) | 250 | 56 | 32 | Train |
| HellaSwag (Zellers et al., 2019) | 1000 | 10003 | 20 | Train |
| StoryCloze (Mostafazadeh et al., 2017) | 1000 | 1871 | 70 | Train |
| WinoGrande (Sakaguchi et al., 2021) | 1000 | 1767 | 50 | Train |
| WSC (Levesque et al., 2011) | 554 | 104 | 32 | Train |
| COPA (Roemmele et al., 2011) | 400 | 100 | 32 | Train |
| WiC (Pilehvar & Camacho-Collados, 2019) | 1000 | 638 | 32 | Train |

Table 3: Size of splits used for experiments across datasets. '#Shots' indicates the number of shots used in few-shot experiments, and 'retrieval from' indicates which split we selected retrieval data from. * Following SuperGLUE Wang et al. (2019), RTE data is from RTE 1/2/3/5 (Dagan et al., 2006; Bar Haim et al., 2006; Giampiccolo et al., 2007; Bentivogli et al., 2009)

## A  DATASET DETAILS

**Sizes and Splits**  For each dataset used, we provide the number of retrieval and validation examples used in Table 3. We also indicate if the retrieval data was split from the validation or training split. Note any data used to retrieve is held out of the validation split to avoid information leakage. We additionally provide the number of shots used for each dataset. We follow the number of splits used by Liu et al. (2022) and use the data shared by the authors (available at https://github.com/r-three/t-few/tree/master/data/few_shot).

**Prompts**  We list the prompts used for each dataset. {x} indicates a space that is filled in by instance data.

- **CaseHold**: What is the correct holding statement for the following text? Text: {context} (A): {ending 1} (B): {ending 2} (C): {ending 3} (D): {ending 4} (E): {ending 5}

- **DROP**: Passage: {passage} Question: {question} Answer:

- **QasperEvidence**: Question: {question} Paragraph: {paragraph} Is the answer to the question in the paragraph? Answer Yes or No.

- **RTE**: {premise} Question: Does this imply that "{hypothesis}"? Yes or no?

- **ANLI**: {premise} Question: {hypothesis} True, False, or Neither?

- **CB**: {premise} Question: {hypothesis} True, False, or Neither?

- **HellaSwag**: Complete the description with an appropriate ending: First, {context a} Then, {context b} ... (a) {ending 1} (b) {ending 2} (c) {ending 3} (d) {ending 4}

- **StoryCloze**: {input sentence 1} {input sentence 2} {input sentence 3} {input sentence 4} What is a possible continuation for the story given the following options ? - {answer 1} - {answer 2}

- **WinoGrande**: {sentence} What does the _ in the above sentence refer to? {option1} or {option2}?

- **WSC**: Passage: {text} Question: In the passage above, does the pronoun '{span 1}' refer to '{span 2}'? Answer:

- **COPA**: {premise} As a consequence... Help me pick the more plausible option: - {choice 1} - {choice 2}

- **WiC**: {sentence 1} {sentence 2} Question: Is the word '{word}' used in the same sense in the two sentences above? Yes, No?

| Task | DEFT-Few (20-70Q) |
|---|---|
| RTE | $73.2_{4.0}$ |
| ANLI R1 | $36.1_{3.0}$ |
| ANLI R2 | $34.1_{0.9}$ |
| ANLI R3 | $40.6_{2.0}$ |
| CB | $58.2_{10.5}$ |
| HellaSwag | $34.1_{0.7}$ |
| StoryCloze | $95.1_{0.3}$ |
| WinoGrande | $50.6_{1.2}$ |
| WSC | $51.0_{5.1}$ |
| COPA | $87.8_{1.1}$ |
| WiC | $50.8_{1.7}$ |
| Average | 55.6 |

Table 4: Performance of XL size models trained using DEFT with few-shot queries. We report the mean and standard deviation over 5 runs.

| Train Model Size | Base | | XL | |
|---|---|---|---|---|
| Index Model Size | Base | XL | Base | XL |
| CaseHold | 14.8 | **15.8** | 32.6 | **37.2** |
| DROP | 20.8 | **21.3** | 30.4 | **31.0** |
| Qasper | 15.7 | **18.0** | 23.3 | **28.5** |
| RTE | 53.4 | **61.7** | **77.3** | 74.0 |
| ANLI R1 | **33.3** | **33.3** | 39.5 | **39.8** |
| ANLI R2 | **33.4** | 32.8 | 35.3 | **37.5** |
| ANLI R3 | 33.2 | **33.3** | **42.5** | 41.4 |
| CB | **50.0** | **50.0** | **75.0** | 60.7 |
| HellaSwag | 26.0 | **27.9** | 31.7 | **33.1** |
| StoryCloze | 74.0 | **76.8** | 94.4 | **95.3** |
| WinoGrande | 49.5 | **50.4** | **51.4** | 50.6 |
| WSC | 41.4 | **42.3** | **43.3** | 39.4 |
| COPA | **63.0** | 60.0 | 85.0 | **95.0** |
| WiC | **48.8** | 48.3 | 49.5 | **54.9** |
| Average | 39.8 | **42.8** | 50.8 | **51.3** |

Table 5: Performance of DEFT models trained on cross-task neighbors retrieved using different-size models.

## B  FEW-SHOT RESULTS WITHOUT IA3

For 'DEFT-Few (20-70Q)' in Table 2, we trained 5 models using DEFT (as we used 5 few-shot sets per dataset). In Table 4 we report the performance of these models *without IA3 training*. Note we did not train few-shot models for CaseHold, QasperEvidence, or DROP, and so do not report results on these datasets. Notably, RTE, CB, and WSC all have quite large standard deviation ($> 3.0$), which suggests our improvements (or deterioration, for WSC) over T0-3B for these datasets may not be significant.

## C  INDEX MODEL SIZE EXPERIMENTS

We explored mismatching the index model sizes, training XL size models on cross-task neighbor splits indexed and retrieved using T5-base, and vice-versa. We use a query size of 1000 and retrieve 500 neighbors per query instance. We present the results in Table 5.

| Evaluation Category | DEFT-XL (instruct) | DEFT-XL (no instruct) | Rand-XL | Tk-instruct |
|---|---|---|---|---|
| Answerability | 48.0 | 48.0 | **49.0** | 47.0 |
| Cause Effect Classification | 83.3 | 83.3 | 84.7 | **87.7** |
| Coreference | 61.0 | 51.0 | 43.0 | **83.0** |
| Data to Text | 34.0 | 34.4 | 33.4 | **37.9** |
| Dialogue Act Recognition | 65.0 | 61.0 | 59.0 | **68.0** |
| Entailment | 50.0 | **68.0** | 13.0 | 19.0 |
| Grammar Error Correction | **86.3** | 84.8 | 84.7 | 84.8 |
| Keyword Tagging | 17.4 | 17.6 | **19.2** | 13.3 |
| Overlap | 17.7 | 20.2 | **22.3** | 17.8 |
| Question Rewriting | 45.8 | 64.0 | 59.9 | **68.8** |
| Title Generation | **21.4** | 20.9 | 20.3 | 20.4 |
| Word Analogy | 60.0 | 41.0 | 60.0 | **61.3** |
| **Average** | 49.2 | 49.5 | 45.7 | **50.7** |

Table 6: Performance of XL-size models on 12 tasks from evaluation categories in Wang et al. (2022). All results are in RougeL. 'Instruct' and 'no instruct' variants of DEFT-XL refer to models trained using subsets of Super-Natural Instructions that were retrieved using instructions and without using instruction respectively.

## D  SUPER-NATURAL INSTRUCTIONS EXPERIMENTS

We use version 2.7 of the Super-Natural Instructions dataset and use the official splits provided, with 100 samples per train and evaluation tasks. This results in a pool of 75,317 train examples. For evaluation, we randomly select one task per evaluation category in Table 5 of Wang et al. (2022). Task names are given in Table 7. We then generate **two** indices for retrieval: one where each sample is encoded including the task instruction, and one where each sample is encoded without any instruction. We then retrieve using the 100 unlabelled test instances from each chosen evaluation task, matching the format used for the index (i.e., if we retrieve from the index with instructions, we encode our query data with instructions included). In order to isolate the effect of instructions on retrieval, after retrieving examples, we always train on the corresponding examples with instructions included (i.e., when we retrieve examples without using instructions, we add the instructions back into the inputs before finetuning). On average, we retrieve 3.5k training examples, roughly 5% of the total training data. Additionally, we finetune a T5-XL model using all available training data ('Tk-instruct'), and a random baseline using random subsets of the training data of the same size as the retrieved subsets ('Rand-XL').

We present our results in Table 6. We find that the instruction-augmented and no-instruction retrieval DEFT models achieve similar performance on average, although the no-instruction variant performs slightly higher. Both DEFT models significantly outperform the Rand-XL baseline, suggesting that the retrieval is still effective even when using a large pool of multitask data without instructions or prompts. However, we find that neither DEFT model significantly outperforms Tk-instruct, which we hypothesise is related to the significantly smaller size of Super-Natural Instructions compared to P3. However, we note that our DEFT-XL models are trained on significantly less data than Tk-instruct, and training all 12 DEFT models is still cheaper than training the Tk-instruct model, using roughly 42,000 examples overall, roughly 56% of the data used to train Tk-instruct.

## E  RETRIEVED EXAMPLES

For a single query from each dataset, we present the top two closest datapoints retrieved below. **Content warning: some of these datapoints reference sensitive topics.** Queries are chosen randomly. Answers are in *italics*.

| Evaluation Category | Task |
|---|---|
| Answerability | `task020_mctaco_answerability_classification` |
| Cause Effect Classification | `task391_cod3s_cause_effect_classification` |
| Coreference | `task1391_winogrande_coreference_resolution` |
| Data to Text | `task957_e2e_data_to_text` |
| Dialogue Act Recognition | `task879_schema_guided_dstc8_dialogue_act_recognition` |
| Entailment | `task937_defeasible_nli_atomic_textual_entailment` |
| Grammar Error Correction | `task1557_jfleg_grammar_error_correction` |
| Keyword Tagging | `task613_liar_keyword_tagging` |
| Overlap | `task039_qasc_overlap_extraction` |
| Question Rewriting | `task670_ambigqa_question_rewriting` |
| Title Generation | `task1356_xlsum_title_generation` |
| Word Analogy | `task1155_bard_word_analogy` |

Table 7: List of tasks used for each evaluation category given in Table 6.

---

**RTE**

**Query:** Thanks to a global ban on the ivory trade that was passed in 1989 by the Convention on International Trade in Endangered Species of Wild Fauna and Flora (CITES), the African elephant population may be reversing its spiral toward extinction\n Question: Does this imply that "The ban on ivory trade has been effective in protecting the elephant from extinction."? Yes or no?

**Retrieved #1:** Title: Dissappointed\n Review: The software works OK, but haven't gotten any more than three numbers on a draw six lottery after 8 months of trying. The biggest thing to watch out for is support, or lack of. If you rebuild your computer or buy a new one and have to re-install their software, you have to get another product ID from them. It took me almost two weeks of begging and a phone call (just an answering machine on their end) to get a response from them. I am coming up on a week of trying to get a response from them for a product ID for my new computer. Funny, because they responded the next day when I first baught the program and they had my money in hand!\n Does this product review convey a negative or positive sentiment? *Negative*

**Retrieved #2:** You are considering whether to buy a product. You look at the reviews. Would the following review decrease or increase the chances of you buying the product?\n Review title: Amazon Rip Off\n Product review: What a huge waste of money. I paid $$$ on this very site not but a month ago, now it is $$. Got it home, followed the instructions and the silly thing will not get but about a foot off the ground if that, and then it just falls over and beats itself into the ground. Don't waste your cash on this, give your kid a fifty dollar bill and let them light it on fire, they'll have for fun. *decrease*

---

**ANLI R1**

**Query:** Secrets of the Cryptkeeper's Haunted House was a childreńs Saturday-morning game show that ran on CBS. It premiered on September 14, 1996 and lasted until August 23, 1997. It featured the Cryptkeeper of "Tales from the Crypt" (with John Kassir as the voice) now serving as an announcer. It is the last TV series in the "Tales From the Crypt" franchise.\n Question: The Secrets of the Crypt Keepeŕs House television show aired on CBS until 1997, and then was picked up and aired on NBC for an additional season. True, False, or Neither?

**Retrieved #1:** Is there a negative or positive tone to this product review?\n ===\n Title: Not quite as good as some others\n Review: This is a fair book, but it is not near as good as Peter O. Steiner's "Thursday Night Poker." Andy Nelson's book can't decide whether it is for beginners or advanced, so it tries to fit advanced technique into too short of space. It barely scratches the surface of any of the topics it brings up. When it doesn't do that, it simply says, "Play so tight that you don't even have to think. Fold 99% of your hands." That does not make for a fun night, in my opinion.\n Answer: *Negative*

**Retrieved #2:** a delegation from the islamic resistance movement -lrb- hamas -rrb- left the gaza strip monday morning , heading for egypt to hear israel 's response regarding a cairo - mediated ceasefire . In a nutshell, *hamas leaders leave to cairo for final ceasefire discussions*

**ANLI R2**

**Query:** The Sea Wall (French: Un barrage contre le Pacifique ) is a 2008 film by Cambodian director Rithy Panh in a French/Cambodian/Belgian co-production. The film opened on 7 January 2009 in France. It was adapted from the 1950 novel "The Sea Wall" by Marguerite Duras. The novel had previously been adapted as "This Angry Age" by René Clément in 1958.\n Question: Marguerite Duras directed the film. True, False, or Neither?

**Retrieved #1:** Title: Exactly what I had been looking for!\n Review: I've gone through two other iPod FM transmitters that I ended up giving away because the quality was less than desirable. After seeing this one pop up in my Quick Picks last week I decided to give it a try. I used it the very first evening I received it and I'm happy to say my search is over. As others noted, use a low FM frequency for the best results (87.9 in my area works well). I don't receive any interference and the music on my iPod comes through just like I expected. For the price, this is definitely the best deal out there.\n Is this product review negative? *No*

**Retrieved #2:** Based on this review, would the user recommend this product?\n===\n Review: My friend tried to commit suicide, and while he was bleeding to death, he was watching mtv, and the video for "Hold On" was playing, and he was like "yeah" and after he was done rocking out he got all inspired and called for an ambulance. And now he's still here, and he takes pills that make him tired, and everyone is careful to be very nice to him and be his best friend, even though we all secretly pity him. Thank you so much.\n Answer: *No*

**ANLI R3**

**Query:** Well, I think during the campaign, particularly now during this difficult period, we ought to be speaking with one voice, and I appreciate the way the administration has worked hard to calm the tensions. Like the vice president, I call on Chairman Arafat to have his people pull back to make the peace.\n Question: Chairman Arafat needs to pull back his people during this difficult time. True, False, or Neither?

**Retrieved #1:** Title: clinton pushes for greater diversity on wall street\n\n===\n\n Write an article with the given title: *u.s. president bill clinton urged wall street brokers to pursue business in america 's economically distressed cities , saying it 's an untapped market with more buying power than mexico .*

**Retrieved #2:** You are considering whether to buy a product. You look at the reviews. Would the following review decrease or increase the chances of you buying the product?\n Review title: Mistake\n Product review: I didn't want to "purchase" Bars and Tones". It was a mistake to click on it. This review doesn't deserve so many words.\n *decrease*

**WiC**

**Query:** It may rain in which case the picnic will be canceled.\n A window case.\n Question: Is the word 'case' used in the same sense in the two sentences above? Yes, No?

**Retrieved #1:** Title: remains of  exhumed from mass graves in eastern croatia\n \n===\n \n Write an article with the given title: *thirty bodies believed to be croats killed by ethnic serbs at the outbreak of the - serbo-croatian war in former yugoslavia have been exhumed from two mass graves in eastern croatia , an official said tuesday .*

**Retrieved #2:** You are considering whether to buy a product. You look at the reviews. Would the following review decrease or increase the chances of you buying the product?\n Review title: For the 50-cent table\n Product review: My favorite author has run out of steam! His co-author does not, repete, does not have the Paterson style. After sampling this "tandemly"-wriiten book, it becomes obvious that this is a time-waster. Even the editing is bad. I didn't feel guilty about not finishing it. It's headed for the community library's monthly book sale–fifty cent table.\n *decrease*

**COPA**

**Query:** The woman filed a restraining order against the man. As a consequence... \n Help me pick the more plausible option:\n- The man called her.\n- The man stalked her.

**Retrieved #1:** First sentence of the article: when christopher darden got a recent early-morning call from his publisher that his book " in contempt " had become no. on the new york times best-seller list , he mumbled something like " ok , " then rolled over and went back to sleep .\n\n Title: *contempt does n't fill christopher darden*

**Retrieved #2:** "Extract the answer to the following question from the movie plot. If the question isn't answerable, please output "Can't answer".\n Question: Who is the toy's leader and Andy's favorite toy?\n Title: Toy Story\n Movie plot: A boy called Andy Davis (voice: John Morris) uses his toys to act out a bank robbery. The bank is a cardboard box, the robber is Mr. Potato Head (voice: Don Rickles) assisted by Slinky Dog (voice: Jim Varney), and the bystanders include Bo Peep (voice: Annie Potts) and her sheep. The day is saved by cowboy doll Woody (voice: Tom Hanks) playing the sheriff, with help from Rex the dinosaur (voice: Wallace Shawn). Woody is the only toy who gets to say his own lines because he has a pull-string that makes him say things like "Reach for the sky!" and "You're my favorite deputy!"During the opening credits (soundtrack: Randy Newman's "You've Got a Friend in Me"), Andy takes Woody downstairs to find his mother (voice: Laurie Metcalf) decorating the dining room for his birthday party. He asks if they can leave the decorations up until they move, and his mom agrees. She says the guests will arrive soon and sends him back upstairs to get his baby sister Molly (voice: Hannah Unkrich), whose crib is in his room. Andy tosses Woody onto his bed before he pulls Molly out of her crib and carries her away.Woody and the other toys have seemed limp and inanimate up to this point, but as soon as Andy leaves the room, Woody sits up and expresses surprise that the birthday party is today. He calls "Ok, everybody, the coast is clear," and the other toys come to life too. Woody calls a staff meeting and tells Slinky Dog to spread the word. Within a few minutes (during which Bo Peep makes a date with Woody for that evening), all the toys are assembled. Woody starts by reminding them all to find a moving buddy so they don't get lost when the Davis family moves to their new house, which will happen in a week. Then he tries to downplay the news that Andy's birthday party is happening today, but it causes a commotion as the toys know that Andy's actual birthday isn't till next week. Rex worries that someone will give Andy another dinosaur, and many of the toys have similar concerns. Woody points out that it makes sense to have the party...\n *Woody*

**WSC**

**Query:** Passage: Dan took the rear seat while Bill claimed the front because his "Dibs!" was quicker. \n Question: In the passage above, does the pronoun "his" refer to Dan?\n Answer:

**Retrieved #1:** Title: I want to READ it on my Kindle\n Review: Why can't I get the read-able version of night for my kindle? I don't want the auidio version...Help! I downloaded it thinking that I would have the choice to read it or to listen to it but that was not the case at all. I'm extremely disappointed.\n Does this product review convey a negative or positive sentiment? *Negative*

**Retrieved #2:** You are considering whether to buy a product. You look at the reviews. Would the following review decrease or increase the chances of you buying the product?\n Review title: Look weird - feel great!\n Product review: These look so weird and also feel weird when you first put them on but they are so much fun. I love them for my yoga class, and sometimes wear them at night watching TV because the separation they give your toes is good for your feet overall. Try them... you'll become a fan too!\n *increase*

---

**WinoGrande**

**Query:** The phone of Donald is a lot better than Adam's because _ paid extra for his phone.\n What does the _ in the above sentence refer to? Donald or Adam?

**Retrieved #1:** Title: more than you expect\n Product review: The thing about these tillers is that they do things you might not think about. For instance, they're great for dealing with long-rooted weeds. You can hack your way down to the root, then pull up the plant and not leave a huge hole in the ground.\n Would you say this review depicts the product in a flattering or unflattering light?\n *flattering*

**Retrieved #2:** Title: purported statement from al-qaida-linked group says ultimatum against italy ends threatens attacks\n\n===\n\n Write an article with the given title: *a statement released sunday in the name of an al-qaida-linked group said the italian government has " dug its grave by its own hands " after it ignored a warning to withdraw its troops from iraq by aug. .*

---

**HellaSwag**

**Query:** Complete the description with an appropriate ending:\n First, [header] How to make a butterfly out of plastic spoons [title] Gather the materials you will need for this project, listed below. [title] Put a craft cloth or some newspaper down on your working surface. [title] Cut the top portion of the four spoons off (leaving about half an inch of the handle left. Then, ...\n(a) [step] ) the top portion is the flat spoon part. [title] Using the cut-off handle of one of the spoons, make that the " body " of the butterfly.\n(b) [step] As the buttons stretch they will process the solution. ) [title] Put one of the red plastic buttons (left over for the butterfly) on top of the red plastic spoon.\n(c) [step] And when done, you should have 8 small, square, circular spoons. [title] Use these four spoons as either a mirror, or a butterfly.\n(d) [step] ) [title] Thread a thin needle into a cast iron ring with four ends attached. [step] Thread the needle to the ring, and then sew the top portion onto the handle of the spoon at the top, or the side which should be facing away from the spoon.

**Retrieved #1:** Title: hmm...\n Review: I bought this costume in hopes of wearing for Halloween ( last year). I had even separately purchased the duster ( which I am now using to really dust things). Uhh... I tried it on ( I got a X-Small) and its just big... the net piece ( part of the dress with the dots) go all the way down to almost my knees. Which makes it awkward and not sexy at all- its just weird I tried tucking the net part in to my undies to hold it, but it just becomes supper puffy-again looks weird. I never wore it and its still brand new sitting in my closet somewhere.Maybe its just for my body- I am not sure, but the material isn't as great either compared to the picture. Def. does not look anything close to how the model looks in it.Sorry- this was not a good buy at all. The model sure looks good in it.\n Does this product review convey a negative or positive sentiment? *Negative*

**Retrieved #2:** What type of details about adolf heeb\n can be gathered from the following bio?\n\n Bio: adolf heeb -lrb- born 11 july 1940 -rrb- is a former cyclist and politician from liechtenstein .\n he competed in the individual road race at the 1960 summer olympics .\n he later served as a member of the landtag of liechtenstein and leader of the patriotic union party .\n \n\n\n- updated \n\n\n\n\n- birth date \n\n\n\n- name \n\n\n\n- birth place \n\n

---

**CB**

**Query:** B: boy, he's a big one. A: he's pretty big. That's why it really surprises me, you know, that he hasn't come back, because, like I said, he's never gone away like this before, and, I would think, you know, I mean, he might could get hurt by a car or something. I don't know that he could really get killed that easily because he is so big.\n Question: he could really get killed that easily True, False, or Neither?

**Retrieved #1:** Summarize this document: Glen Water Limited also paid costs of \u00a31,600 to restore fish stocks in the Tall River near Richhill.\n About 250 metres of the river was affected when untreated sewage was discharged into it.\n It caused what was described as a m̈oderatef̈ish kill.\n Inspectors found a plume of untreated sewage coming from a discharge pipe at Richhill waste water treatment works in 2014.\n An investigation found that an üninterruptable power sourceät the plant had failed.\n In addition, a power cut to the alarm system meant staff were unaware of the problem.\n Glen Water Limited is based at Dartford in Kent.\n Under a 25-year public private partnership it has the contract for 25% of Northern Ireland's waste water treatment capacity.operates and maintains nine treatment works or pumping stations up to 2032 in return for monthly payments.\n Summary: *A company which treats sewage for NI Water under a public private partnership contract has been fined \u00a32,500 for polluting a County Armagh river.*

**Retrieved #2:** Title: Good\n Review: Well, I'd say all of these songs are well constructed, dope lyrics whatever... but wth? all the basslines sound the same or what? Personally i prefer Violent By Design over this.\n Is this product review negative? *No*

---

**StoryCloze**

**Query:** Andy had always wanted a big kids bike. When he turned six Year's old he asked for a bike for his birthday. He did not know how to ride a bike. On Andy's birthday his mother gave him a bike. What is a possible continuation for the story given the following options ?\n - Andy cried for hours.\n - His dad taught him how to ride it.

**Retrieved #1:** Based on this review, would the user recommend this product?\n ===\n Review: I love most Neil Young but every fan knows that about one in three of his albums really sucks. After Greendale and Greatest hits, I'm very disapointed.\n Answer: *No*

**Retrieved #2:** hong kong share prices rose a mere . percent on late overseas buying thursday despite early profit-taking , dealers said .\n \n ===\n \n Given the above sentence, write its title: *hong kong shares close . percent firmer*

---

**CaseHOLD**

**Query:** What is the correct holding statement for the following text?\n Text: component of the res judicata doctrine. The Ohio Supreme Court held that the original criminal proceedings in Krahn were insufficient to invoke collateral estoppel in the later malpractice case because the claimed error by Krahn's criminal lawyer in plea negotiations was not " 'actually and necessarily litigated and determined' in the denial of her motion to vacate the criminal judgment against her." Krahn, 43 Ohio St.3d at 108, 538 N.E.2d 1058, quoting Goodson v. McDonough Power Equip., Inc. (1983), 2 Ohio St.3d 193, 195, 2 OBR 732, 443 N.E.2d 978. The Supreme Court by no means suggested that collateral estoppel was completely inapplicable in the context of a criminal conviction when, as here, matters genuinely were litigated and determined. Id. at 107, 538 N.E.2d 1058 (¡HOLDING¿). Decisions in Ohio other than Krahn relative \n (A): recognizing the doctrine of collateral estoppel in agency proceedings\n (B): holding that the facts prevent the invocation of collateral estoppel as a bar to krahns cause of action in this case\n (C): holding collateral estoppel elements met considering changed circumstances in the context of an exception to the general rule of collateral estoppel\n (D): recognizing the cause of action\n (E): holding that collateral estoppel applies to 1983 claims

**Retrieved #1:** Is there a negative or positive tone to this product review?\n ===\n Title: Too steep\n Review: I bought this for my dog who had back problems, it was way too steep and my dog had to jump about 3/4's of the way up to my bed because the measurement of the ramp on the description was incorrect. It totally defeated the purpose of my dog having to not jump. I had to go back to the stairs I had been using\n Answer: *Negative*
**Retrieved #2:** Write a title for this sentence: the fate of president barack obama 's top domestic priority – a remake of the u.s. health care system – now rests in the hands of a pivotal but deeply divided senate committee . \n \n Title: *toughest test coming up for health care overhaul*

---

**DROP**

**Query:** Passage: Coming off their overtime win at San Diego, the Broncos traveled to the Mall of America Field at the Hubert H. Humphrey Metrodome for an interconference duel with the Minnesota Vikings. The game's first points came from the Vikings, when defensive end Jared Allen tackled running back Willis McGahee in the end zone for a safety. The Broncos grabbed the lead when linebacker Mario Haggan returned an interception off Vikings' quarterback Christian Ponder 16 yards for a touchdown. Vikings' kicker Ryan Longwell made a 40-yard field goal, then the Vikings reclaimed the lead in the second quarter, on a 19-yard touchdown pass from Ponder to tight end Kyle Rudolph, followed by another field goal by Longwell, this time from 25 yards, just before halftime. The Broncos narrowed Minnesota's lead in the third quarter, with a 21-yard touchdown pass from quarterback Tim Tebow to wide receiver Demaryius Thomas, but the Vikings countered with a 52-yard touchdown pass from Ponder to wide receiver Percy Harvin. The Broncos responded with a 41-yard touchdown pass from Tebow to Thomas. Ponder then threw a 48-yard touchdown pass to Harvin to give Minnesota a 29-21 lead. On the Broncos' next possession, McGahee rushed 24 yards for a touchdown and Tebow scrambled for a two-point conversion to tie the game at 29. The Vikings subsequently reclaimed the lead on Longwell's 39-yard field goal with 3:06 left in the game. The Broncos answered with kicker Matt Prater's 46-yard field goal with 1:33 left to tie the game at 32. On the Vikings' ensuing possession, Broncos' cornerback Andr233; Goodman returned an interception off Ponder to the Vikings' 15-yard line. Six plays later, Prater nailed the game-winning 23-yard field goal as time expired to give the Broncos their fifth consecutive win.\n Question: how many yards did longwell make?\n Answer:

**Retrieved #1:** Make a title for this article: andy roddick hit a record-breaking  mph -lrb- . kph -rrb- serve friday in a lopsided win over stefan koubek as the united states took a - davis cup lead over austria . \n \n *roddick ginepri give united states - lead over austria*
**Retrieved #2:** Orton does not start against Ohio State Purdue quarterback Kyle Orton did not start Saturday 39;s game against Ohio State, though he was listed as available to play. Orton has been bothered by a right hip injury for the last month. \n \n Which of the following sections of a newspaper would this article likely appear in? World News, Sports, Business, or Science and Technology? *Sports*

**Qasper**

**Query:** Question: How big is Augmented LibriSpeech dataset? Paragraph: We introduce a multilingual speech-to-text translation corpus, CoVoST, for 11 languages into English, diversified with over 11,000 speakers and over 60 accents. We also provide baseline results, including, to our knowledge, the first end-to-end many-to-one multilingual model for spoken language translation. CoVoST is free to use with a CC0 license, and the additional Tatoeba evaluation samples are also CC-licensed. Is the answer to the question in the paragraph? Answer Yes or No.

**Retrieved #1:** Title: make your july celebration sizzle\n \n ===\n \n Write an article with the given title: *you have less than a week to get your fourth of july cookout menu set and we thought we 'd help* .

**Retrieved #2:** Title: A good idea...\n Review: that went terribly bad. I cannot comprehend how some of these "artists" were chosen for this. "Atlantic City" and "State Trooper" are embarrasing to say the least, but they sadly showcase what is now Nashville's finest. If Johnny Cash and Dar Williams recordings had not appeared on this CD, one star would have been too many. Thankfully, these mostly pathetic renderings cannot tarnish the greatness of Mr. Springsteen or his amazing album. Go get the original. You won't be sorry.\n Does this product review convey a negative or positive sentiment? *Negative*

