# OpenReview forum: "Data-Efficient Finetuning Using Cross-Task Nearest Neighbors"
_ICLR.cc/2023/Conference — Submitted to ICLR 2023_

### Official Review · Reviewer_fvhK · 2022-10-24

**Confidence:** 5
**Correctness:** 3
**Technical Novelty And Significance:** 3
**Empirical Novelty And Significance:** 4
**Recommendation:** 6

**Clarity, Quality, Novelty And Reproducibility:**

The paper is clear, but I doubt it can be easily reproduced by a third party given the scale of data involved. I recommend the authors release their code.

**Strength And Weaknesses:**

### Strength
1. This paper proposes a simple yet effective idea on how to use existing resources (i.e., P3) to facilitate unseen tasks.
2. The paper is well-written and easy to follow.
3. The evaluation is good - it considers both 0-shot and few-shot settings.

### Weaknesses
1. The idea is not completely novel. Similar approaches exist like kNN-LM. Also, the proposed approach is also related to Mixture-of-Experts.
2. It is unknown if the task retrieval mechanism in the paper is optimal. For example, what if we use BM25 for retrieval? Also is task embedding relevant here?
3. The proposed method has its drawback. It requires retrieval and training for each task. In other words, for 3 tasks there are 3 models trained instead of the open-box T0. In Table 1, DEFT outperforms T0 but the improvement is somewhat limited.

**Summary Of The Paper:**

This paper proposes a new approach to multitask learning by retrieving relevant training examples from P3 and train the model on it.

**Summary Of The Review:**

This paper explores a simple yet effective way to facilitate multitask data but may lack novelty and practicality.

---

> ### Author Response · Authors · 2022-11-17
> **Response to Reviewer fvhK**
>
> We thank the reviewer for noting our method is simple and effective, and that our evaluation settings are good. We plan to release our code shortly to facilitate further research.
>
> *Optimality of task retrieval mechanism*: We have added a BM25 baseline (see table 1) and found that it performs similarly to the random baseline, suggesting that sparse keyword-based approaches can not capture useful similarities for generalization to target tasks. We found in initial experiments that gradient-based methods such as task embeddings [1] were prohibitively expensive for large (> 1 billion parameter) models, but agree that further exploring different retrieval mechanisms is an important and interesting line of research we leave to future work.
>
> *Drawback of proposed method*: We agree that DEFT models are not general multi-taskers, but believe this is a clear tradeoff: you can train multiple models to handle multiple tasks with good performance, or train one model to handle multiple tasks but with lower overall performance. We also note that training all the DEFT models together is still cheaper than training T0 - training all the DEFT models with a batch size of 1024 for 5 epochs takes roughly 5.8k steps, while T0-3B was trained for 11.2k steps with a batch size of 1024, roughly double the compute.
>
> [1] Tu Vu, Tong Wang, Tsendsuren Munkhdalai, Alessandro Sordoni, Adam Trischler, Andrew Mattarella-Micke, Subhransu Maji, and Mohit Iyyer. Exploring and predicting transferability across NLP tasks. In Proceedings of the 2020 Conference on Empirical Methods in Natural Language Processing (EMNLP), pp. 7882–7926, Online, November 2020. Association for Computational Linguistics. doi: 10.18653/v1/2020.emnlp-main.635.
>
> edit: we have edited this response to improve clarity and directly link to the points raised by the reviewer.

---

> > ### Comment · Reviewer_fvhK · 2022-11-23
> > **Thanks for your response**
> >
> > Thanks for your response. I've carefully read it and decided to keep my original score.

---

### Official Review · Reviewer_g3S4 · 2022-10-25

**Confidence:** 5
**Correctness:** 3
**Technical Novelty And Significance:** 2
**Empirical Novelty And Significance:** 2
**Recommendation:** 3

**Clarity, Quality, Novelty And Reproducibility:**

- It would be nice to have Figure 3 and Figure 4 for the few-shot setting, and compare the performance to the T0-3B+IA3 when fine-tuned on the original test task data vs T5-XL-LM fine-tuned on original test task data + 2x, 3x, 4x of augmented data.
- The applicability of the method outside of NLP tasks might be discussed in the conclusion, e.g. in vision or multi-modal settings.

**Strength And Weaknesses:**

Strengths:
- *The approach is simple and effective*: The main idea is simple and effective - perform a "task-attention" on a set of training tasks to augment the test task few-shot data. It is quite interesting to see that a naive example retrieval approach such as taking the union of the retrieved training examples can lead to significant gains (64.6 vs 60.4 on T0 test tasks for the few-shot setting).

- *The paper is clear and well-written*

Weaknesses:

- *Limited novelty w.r.t. related work*: One of the main weaknesses of this paper is the close similarity to ReCross (Li et al., 2022) to appear in NeurIPS 2022, and published on ArXiv last April. ReCross introduces the same idea: retrieving from a set of upstream tasks training data to benefit the downstream task data. The authors acknowledge this similarity in the related works, by stating that their approach differ from ReCross because ReCross "*make use of an additional re-ranker and multi-task trained model to retrieve useful instances.*". Per se, not using a modeling choice of a previously published paper is *not* a significant novelty: the novelty and contribution would come from showing that the modeling choice used in ReCross *is not necessary* or even *harmful* for achieving good performance in a different setting. These conclusions are currently missing from the paper. Given the good scores the paper reports on the T0 suite, I can imagine that the reranker might be only marginally useful, but this should be clearly stated and empirically supported. The community goes fast and probably the authors started their work before having noticed the relevant related work. Nevertheless, at the moment, this paper presents the idea as totally new. It would be more suitable, for example, to acknowledge the similarity to Li et. al, 2022 directly in the intro and maybe make an extra effort to offer a simplification/ablation of some components of the aforementioned approach. Some of the questions I can think of that would give a bit of novelty and require a small additional effort are:
1) Is the reranker used in ReCross useful?
2) What happens if you fine-tune a multi-task trained model (T0-3B in your case, similar to ReCross), rather than a "generalist" model (T5-XL-LM)?

- *Experimental setting can be stronger*: Differently from ReCross, the authors only test on the T0 evaluation suite and build heavily on top of IA3 (Liu et al, 2022). I do believe that another benchmark is necessary. It would be interesting for example to explore the proposed method with Natural Instructions V2 dataset. For example, this would give a better idea if the method works due to the nature of the prompts format in P3 (e.g . it retrieves examples with similar prompts) or due to other mechanisms.

- *Claims about task interference*: The authors claim that their result show that there is negative task interference. I would think that an important experiment to substantiate this claim is to report the results for 2) above, where the method is applied on top of a multi-task model trained on the multi-task training set.

**Summary Of The Paper:**

The paper proposes a simple data-efficient fine-tuning technique, where the few-shot data for a test task is augmented by retrieving similar examples from a multi-task training set (which does not contain the exact test task). The test task training set is augmented by taking the union of the training examples retrieved for each example of the test task. The authors operate in a setting where only unlabeled data for a test task is available to do retrieval and a setting where also labeled data is available. The method demonstrates gains on the test datasets used to evaluate T0, when compared to multi-task baselines that are either applied zero-shot to test tasks (e.g., T0) or are fine-tuned on the few-shot test task data (e.g., T-Few).


**Summary Of The Review:**

This paper proposes a simple and effective idea for improving zero-shot / few-shot task generalization. The reported results are convincing. Unfortunately, the overall contribution of this paper is diminished by the limited novelty with respect to ReCross (to appear at NeurIPS, 2022, and published on arXiv in April). The authors cite this paper in the related work, but I feel that the similarity is quite understated. It is questionable whether a paper published in April should be considered related work or not and whether this should impact my review. In the current context, I would suggest the authors to do a small extra effort to 1) highlight similarities with ReCross and 2) differentiate their contribution w.r.t. ReCross, for example, by reporting ReCross performance and ablating some of the ReCross components (e.g. the reranker and the fact that ReCross uses the multi-task model as base model). Right now, the paper appears to be written as if it was the first in proposing this retrieval idea. I didn't discover the similarity until I read the ReCross work in details. All in all, I find the idea exciting: the strong similarity to a previous work, despite this being quite recently published online, requires the paper to offer new perspectives that are not present just yet; experimental setting could be made stronger by adding an additional dataset (e.g. NIv2), to show more convincingly which aspects of the method are responsible for improved performance.

---

> ### Author Response · Authors · 2022-11-17
> **Response to Reviewer g3S4 - Part 1/2**
>
> We thank the reviewer for their insightful and useful comments, which have helped us to greatly improve the paper.
>
> *Comparison with ReCross*: We agree that the distinction between our setting and ReCross can be made clearer and have updated the paper to include more explicit comparisons and note where our methods are similar, with updates primarily made in the introduction (when introducing the idea of cross-task neighbors, citing ReCross) and discussion (adding the results reported by ReCross and comparing to DEFT, see section 4.2). **We note that according to ICLR guidelines, ReCross is considered concurrent work as it is being published at NeurIPS 2022 (starts Nov 28 2022)**. In addition, we believe that our work is significantly different to ReCross, with the similarity limited to the method used for identifying relevant source task data. We list some of the differences below:
> - We focus on **pruning the multi-task training data to only relevant subsets** while ReCross **augments the multi-task training**. As such, we report findings comparing models trained on data subsets with models trained on random subsets/all data, while ReCross compares models trained on available data together with small amounts of retrieved data. Our overall aim is to make training models **more data efficient**, while ReCross aims to **augment performance**, and **this is reflected in the significantly different findings reported and claims made in our work**.
> - **We do not assume access to multi-task trained models** for indexing and retrieval, instead using general pretrained language models. In fact, while ReCross claims that using a multi-task model is important for the retrieval to work well, **we show strong results using simple pretrained language models** at XL (3B parameter) and base (~250 mill parameter) sizes.
> - **DEFT models outperform ReCross** on 4/6 tasks reported by the ReCross authors, despite ReCross training on a large amount of P3 training data (1.7 million examples) and then further training on retrieved data.
> - We experiment with far larger query sets (1000 examples instead of ~16), although our approach works with smaller sets as well.
> - We retrieve far larger retrieval-selected subsets (thousands instead of hundreds of examples).
>
> We believe that our significantly different motivation yields novel insights and perspectives on this method, and that our paper showcases a novel use-case for cross-task retrieval in showing how it can be used to prune, rather than augment, training data for multi-task models, yielding much more significant performance and efficiency gains. **The similarity between ReCross and our work is only limited to the method used for identifying relevant source task data** (retrieving nearest neighbors based on the encoder’s representations). We would like to highlight that the method itself is not our primary contribution, and we only chose the simplest method for identifying the relevant data points that does not rely on target task labels. We have noted the similarity in our introduction (second paragraph), results (table 1), and discussion (section 4.2) and believe our work provides a novel perspective on this method.
>
> *Effect of ReCross Reranker*: Unfortunately, the ReCross authors have not made their trained reranker model public, making it difficult to easily ablate in our setting, but we note that the gap between ReCross with and without the re-ranker is quite small using accuracy as a metric (37.01±0.94 vs 37.47±0.73, table 7 of [1]). As such, we have not performed a precise ablation by integrating the ReCross reranker into DEFT, but we do compare the reranker-augmented ReCross with DEFT and find that DEFT outperforms ReCross with reranking (see updated table 1) on 4/6 datasets.
>
> *Effect of using T0-3B in DEFT finetuning*: We first note that the core premise of our work is data-efficient finetuning, reducing the length and cost of the multi-task finetuning stage and enhancing target task generalization. Making use of a trained multi-task model violates this since it requires training that model on a large amount of data, which is more costly than training all DEFT models in table 1. We tested using T0-3B instead of T5-XL as the starting model and training on the data subsets used to train DEFT-XL in table 1 and find this performs slightly better: 52.3 vs 51.3 average performance across tasks (i.e., the setup used in table 1), which suggests the extra multi-task training does aid in making use of the retrieved examples. However, this is not data-efficient.
>
> We continue our response in Part 2/2.
> Edit: we have edited this comment to better address the reviewer's concerns.

---

> > ### Comment · Reviewer_g3S4 · 2022-11-21
> > **"ReCross augments the multi-task training" while "you prune the multi-task training data to only relevant subset" ?**
> >
> > Thank you for your response.
> >
> > I don't understand why you say that "ReCross augments the multi-task training" while you "prune the multi-task training data to only relevant subset"
> >
> > If I look in the ReCross paper, Section 3.1, it states:
> >
> > "The ReCross method is also based on the simple idea that we should exploit the upstream examples that enjoy better utility for a given unseen target task. Instead of costly re-training from scratch, our method first retrieves a small subset of the upstream data for each unseen task. It then uses them to efficiently fine-tune the upstream model such that the updated model is generalized. Ideally, we aim to retrieve the upstream examples that are the most beneficial ones for generalizing the upstream model towards a particular unseen task — ranking the upstream data by their example-level utility."
> >
> > As I see it, for a given target task with some unlabeled data ReCross 1) retrieves similar labeled examples, 2) adapts a multi-task trained model with those labeled examples + unlabeled examples from the target task annotated w distant supervision.
> >
> > The difference of your setup is that 1) you have some labeled data for the target task and 2) you adapt a general model and not a multi-task trained model.
> >
> > From this point of view, ReCross seem a generalization of your approach.
> >
> > Am I missing something?

---

> > > ### Author Response · Authors · 2022-11-23
> > > **Re: Difference between ReCross and DEFT**
> > >
> > > **1.** ReCross and our work are complementary, using a similar method in a different setting. The core difference in our setup is that *we do not assume access to a pretrained multi-task model (e.g., T0), and do not assume access to labels for the target task* (except when exploring further few-shot finetuning). We believe this setting has real-world applications: for example, a practitioner may easily be able to gather a large pool of data (e.g. collating many datasets in huggingface datasets) but *only care about a few downstream tasks*. Training a large multi-task model in this scenario is expensive and costly, but training a handful of DEFT models is much cheaper, as DEFT uses a small fraction of data available in the pool (see our response to comment #3 for a similar but slightly more formal version of this argument). As such, a practitioner in this scenario may choose to use DEFT to instead select a relevant subset of their large pool of data and train a model on this alone, achieving better performance than if they chose a random subset. In contrast, ReCross assumes that the practitioner instead does this costly multi-task training and then wants to further improve performance on some set of downstream tasks. As such, these two works use similar methods to tackle different scenarios.
> > >
> > > We also note that DEFT works *even if the massive pool of data is unlabeled*, as we encode data without the labels. This means that a practitioner could gather a large number of potential examples, find the DEFT subset, and then only label these examples. This is especially relevant considering that tasks like QasperEvidence and CaseHOLD are costly to gather annotations for, as noted in section 5.2, and a practitioner may choose to label the DEFT subset instead of labeling the target task data. ReCross does not apply in this setting, as training the multitask model and reranker requires labeled multitask examples. We admit that this use is not clear in the current work (despite it partially motivating our choice of evaluation datasets) and will make it clearer for the camera-ready submission.
> > >
> > > **2.** ReCross and our work are concurrent. We also note again that ReCross, as work published at NeurIPS, is concurrent work as per ICLR policy (quote below from https://iclr.cc/Conferences/2023/ReviewerGuide):
> > >
> > > >"Q: Are authors expected to cite and compare with very recent work? What about non peer-reviewed (e.g., ArXiv) papers? (updated on 7 November 2022)
> > > >
> > > > A: We consider papers contemporaneous if they are published (available in online proceedings) within the last four months. That means, since our full paper deadline is September 28, if a paper was published (i.e., at a peer-reviewed venue) on or after May 28, 2022, authors are not required to compare their own work to that paper. Authors are encouraged to cite and discuss all relevant papers, but they may be excused for not knowing about papers not published in peer-reviewed conference proceedings or journals, which includes papers exclusively available on arXiv. Reviewers are encouraged to use their own good judgement and, if in doubt, discuss with their area chair."*
> > >
> > > Nonetheless, we agree that it is important to cite and note the similarities and distinctions between ReCross and our work. As previously suggested, we have included this in the related work section of our submission, and have extended the references to ReCross in the most recent version of our paper (adding notes on ReCross in the introduction, extending its discussion in related work, and noting it in the discussion of results).

---

> > ### Comment · Reviewer_g3S4 · 2022-11-21
> > **Super-Natural Instructions**
> >
> > Thank you for your additional experiments on Natural Instructions.
> >
> > In your answer, "Effect of using T0-3B in DEFT finetuning" you state that "However, this is not data-efficient.". I would say that your DEFT model is *no more* data-efficient than pre-training T5-XL on the multi-task training set as you assume that you can *query* all examples in that training set. It is though more *compute efficient*, given that you skip the multi-task pre-training.
> >
> > Given that Tk-Instruct outperforms all other methods in the NI dataset, I wonder why a practitioner would prefer to skip the multi-task pre-training and just use DEFT-XL. DEFT-XL is *less* computationally efficient than Tk-Instruct if one considers that multi-task pre-training needs to be done once, while *retrieval and fine-tuning* needs to be done for each task.
> >
> > It might be worth trying to improve over Tk-Instruct by applying DEFT on top of a Tk-Instruct model. Did you try this?

---

> > > ### Author Response · Authors · 2022-11-23
> > > **Re: Super-Natural Instructions Results**
> > >
> > > *Why a practitioner would prefer to skip the multi-task pre-training and just use DEFT-XL?*
> > >
> > > Let us consider the cost of doing multi-task pretraining versus DEFT-XL given some large pool of data. Massive multitask training costs O(M), where M is the size of the multitask pool. DEFT training costs O(EC), where E is the number of evaluation tasks and C the number of cross-task neighbors retrieved per task. For our experiments, C is often much smaller than M (< 2% for P3 and ~5% for Super-Natural Instructions), and so we believe that there are many values of E for which DEFT is more efficient. Therefore, if a practitioner cares only about a small number of downstream tasks (roughly < 20 for Super-Natural instructions, and roughly < 50 for P3), training DEFT-XL is much cheaper than doing pretraining on the entire multi-task pool. We would like to note that practitioners typically want to build models that do well at a small set of specific target tasks.
> > >
> > > *Applying DEFT on top of Tk-instruct.*
> > >
> > > Given the above motivation and the assumptions listed in point 1 of our comment *Re: Difference between ReCross and DEFT *, applying DEFT on top of Tk-instruct is more aligned with the objective of ReCross. The point of DEFT is not to push a multi-task model to the best possible performance, but instead to allow a practitioner who cares about the performance of a small number of (expensive to label) tasks to prune the multitask dataset and train a model with a much smaller computational cost.

---

> > ### Comment · Reviewer_g3S4 · 2022-11-21
> > **ReCross results in the paper**
> >
> > It might not be fair to compare and claim improvements with the ReCross results reported in the original paper, as the authors use a *BART*-based baseline given that they cannot fit the larger T5-XL model.

---

> > > ### Author Response · Authors · 2022-11-23
> > > **Re: ReCross Results**
> > >
> > > We first note that recreating ReCross in our setup is expensive as creating a new index and large-scale model training takes significant resources. We agree that the reported ReCross results are not directly comparable to DEFT and are happy to either (a) alter the table and caption to note these differences or (b) remove the result entirely. We will make these changes for the camera-ready submission.

---

> ### Author Response · Authors · 2022-11-18
> **Response to Reviewer g3S4 - Part 2/2**
>
> Continued from *Response to Reviewer g3S4 - Part 1/2*:
>
> *Strength of experimental setting*: We note that we test DEFT on **three datasets in addition to the T0 evaluation suite**: DROP, CaseHOLD, and QasperEvidence, which represent tasks quite different to those found in the T0 evaluation suite, and are not found in P3.
> However, we agree that exploring the effect of prompted data is important and update our work with results on SuperNatural Instructions (formerly called Natural Instructions V2) data with and without instructions. We encode instances with and without instructions respectively for retrieval, but train on instances with instructions included in order to precisely test the effect of instructions on retrieval only. We find that retrieving examples without instructions performs slightly better than with instructions, although both models perform similarly overall and outperform a random sample baseline. This suggests that it is the sample information, rather than the prompt, that provides useful information for cross-task retrieval. We also find that DEFT-trained models are on average able to perform similarly to a T5 model trained on all available data (100 training instances for all tasks in the English split of Super-Natural Instructions). We discuss these results in section 5.2 (‘Are prompts useful for retrieval?’) and report detailed results in Appendix D.
>
> *Task interference*: We add a new baseline, Rand-bal, which uniformly samples from all tasks in P3, and underperforms DEFT. The fact that DEFT outperforms Rand-bal, Rand, T0 baselines, which are identical models simply trained on different data, suggests that it is the DEFT data selection that allows better generalization to the target task and that the additional data present in T0 training makes it difficult for the model to generalize well.

---

### Official Review · Reviewer_SJym · 2022-10-26

**Confidence:** 5
**Correctness:** 3
**Technical Novelty And Significance:** 3
**Empirical Novelty And Significance:** 3
**Recommendation:** 8

**Clarity, Quality, Novelty And Reproducibility:**

Questions and comments:
- How did you handle imbalances in the P3 dataset? P3 has large subsets that can be 100x bigger than other subsets. If these imbalances are not properly handled, DEFT could be overly sampling (random) or retrieving from the most represented subsets rather than the most relevant subsets (because the most relevant subsets are potentially small). In term, the “negative transfer” argument becomes less convincing.
While I agree, it’s hard to re-train T0-3B on a well balanced P3, I’d like to see the random sampling baseline with a balanced P3 if it’s not already the case.
- Introduction: “These findings suggest that prior work may have underestimated the cross-task generalization capabilities of large multitask models”. I think this is a generous claim or at least imprecise (if prior works mean T0/FLAN): while previous works have focussed on the true zero-shot case (where there is no knowledge of the target task, nor through its prompt or through any labeled or unlabeled instances), the setup you are interested in requires access to target task (unlabeled or labeled). It seems that a given DEFT model optimized for target task A would perform well on a target task A, but worse on a target task B, whereas T0/FLAN would perform relatively worse than a “specialized” DEFT but better on average.
- Could you showcase a few retrieved examples and their associated query?
- Missing reference: https://arxiv.org/abs/2005.00770


**Strength And Weaknesses:**

Strengths:
- The problem is well posed, the method is clearly explained while being relatively novel.
- Numbers are strong and experiments well conducted and results are interesting.

Weaknesses:
- I have a few doubts about some of the stronger claims in the paper (in particular the negative transfer argument, see questions).
- Few-shot fine-tuning has a long history in NLP. I’d like to see a fairer comparison to stronger baselines. IA3 is particular in the sense that it only fine-tunes a fraction of the parameters. Using a parameter efficient fine-tuning baseline needs a better justification (especially since DEFT-Few seems to be fully fine-tuned).


**Summary Of The Paper:**

This paper is interested in building language models that can generalize to new tasks when there is only unlabeled data for that task. The authors introduce a method called DEFT, where unlabeled instances for the task at hand are used to retrieve the most similar labeled prompted data in a pool of prompted instances of NLP tasks (such as the one used to train T0). The retrieved labeled prompted instances are then used to fine-tune an encoder-decoder language model (more specifically T5 LM), and the model is evaluated on the target task. Experiments show that the resulting DEFT model outperforms T0 at the same model size by a strong margin. When a few labeled data is available for the target task, the DEFT models provide a better initialization for few-shot fine-tuning than T0.

The authors use these data points to highlight that only a subset of the pool of prompted instances is required to achieve high performance on the target task and to highlight negative transfer between tasks when doing massively prompted fine-tuning (i.e. FLAN/T0).


**Summary Of The Review:**

Strong paper with well-executed experiments and a somehow novel story. Numbers are relatively strong.
Some minor doubts and questions about the experiments.
Recommend accept.

---

> ### Author Response · Authors · 2022-11-17
> **Response to Reviewer SJym**
>
> Thank you for noting the strength of our method and the interesting results. We have added some examples of what gets retrieved for different queries in Appendix E and added the missing reference to our related work (under ‘Multi-task Transfer Models’).
>
> *Imbalances in the P3 Dataset*: We first note that T0 itself was trained with a balanced sampling procedure to avoid large datasets dominating, to quote the T0 paper: *"[We] treat any dataset with over 500’000 examples as having 500’000 / num templates examples for the purposes of sampling"* (page 6 of [1]). Correcting for this gives a mixture that follows the distribution given in the rightmost column of figure 2 in our paper, which shows that the sampling is fairly even across datasets (although not uniform). This figure also displays what gets retrieved via DEFT, and we note that across datasets each retrieved mixture is fairly unique and quite different from the overall distribution of datasets in P3. As such, while we did not explicitly balance the sampling in the retrieval process, we believe that the retrieved sets are not especially biased towards the massive sub-datasets in P3. However, we agree that the random baselines are potentially biased towards these datasets and so add a baseline using subsets chosen using uniform sampling across tasks, which we find still underperforms DEFT on 8 datasets.
>
> *Few-shot baselines and fairness*: First, we note that T0-3B+IA3 and DEFT-Few are trained **using identical steps**: (1) an initial full-finetuning multitask training stage using either all P3 data or the DEFT-selected subset, and then (2) further parameter-efficient training using IA3 vectors and few-shot task data. This means the parameter-efficient IA3 training is the same between all models in table 2, and **the only difference between them is the choice of data used in the initial multitask training** (since the few-shot finetuning always uses the same labeled target task data). We have clarified this in our updated paper. Our choice of IA3 as the baseline for few-shot finetuning is based on the strong results reported in [2], which suggest that IA3 outperforms full-finetuning and other parameter-efficient methods in a few-shot setting on the T0 evaluation datasets.
>
> *Cross-task Generalization Claim*: As for the cross-task generalization claim, we believe that our results show a gap between these zero-shot models and models trained on subsets of the multitask training data (i.e., our DEFT models). Ideally one would hope that training on the entire multi-task mixture yields models as least as good as the DEFT models, and the fact that this isn't the case suggests that negative interference in the multi-task mixture is causing the degradation in performance. We have reworded this paragraph at the end of the introduction to focus on this negative interference claim: *“These findings suggest that training these models on all the available multitask prompted data results in negative interference, even in relatively large (3 billion parameter) models.”*
>
> [1] Victor Sanh, Albert Webson, Colin Raffel, Stephen H Bach, Lintang Sutawika, Zaid Alyafeai, Antoine Chaffin, Arnaud Stiegler, Teven Le Scao, Arun Raja, et al. Multitask prompted training enables zero-shot task generalization. arXiv preprint arXiv:2110.08207, 2021
> [2] Haokun Liu, Derek Tam, Mohammed Muqeeth, Jay Mohta, Tenghao Huang, Mohit Bansal, and Colin Raffel. Few-shot parameter-efficient fine-tuning is better and cheaper than in-context learning. arXiv preprint arXiv:2205.05638, 2022.
>
> edit: we have edited this comment to make the links to the points raised by the reviewer clearer, and added an extra point about the few-shot results.

---

> > ### Comment · Reviewer_SJym · 2022-12-07
> > **Thank you for response**
> >
> > Thank you for your responsed. I have read it carefully, along with answers to other reviews and am keeping my score.

---

### Official Review · Reviewer_zbKf · 2022-10-26

**Confidence:** 4
**Correctness:** 3
**Technical Novelty And Significance:** 2
**Empirical Novelty And Significance:** 3
**Recommendation:** 6

**Clarity, Quality, Novelty And Reproducibility:**

- Some of the model naming conventions could be clearer
- The method (nearest neighbor augmented retrieval) isn't novel, though providing a detailed empirical assessment of its performance is a good contribution
- Uses public data for reproducibility, though unclear if code will be shared (though again, the methods are straightforward)

**Strength And Weaknesses:**

Strengths
- Optimizing fine-tuning for improved data efficiency (or more targeted task improvement) is an important research area.
- Nice connections to retrieval augmented language models with corresponding retrieval experiments.

Weaknesses
- The approach is extremely simple. This could also be a strength, but I wonder how much using only a single source (P3) of prompts is biasing results, since many templates used for creating that dataset are very similar across tasks. Expanding results across prompted datasets would be a good series of additional experiments.
- Given the hypotheses around the the role of negative interference in hurting performance (and the resulting benefit of querying more similar examples for fine-tuning), the paper would be much stronger if there were direct experiments exploring this
- Some of the model naming conventions are confusing/inconsistent (DEFT-T5-base vs DEFT-base). It wasn't clear to me what DEFT-base was defined as (vs XL which was the 3B param variants).


**Summary Of The Paper:**

The paper presents a method for selecting subsets of prompted multitask training data for fine-tuning  a language model (Data-Efficient FineTuning, or DEFT). Given a small set of unlabeled task examples, the paper uses nearest neighbors to retrieve a targeted task subset from a larger pool of prompted examples. Using this method to build a more data-efficient training set for fine-tuning, the paper reports improvements using a T5 model an equiv. 3B T0 model (outperforming in 11/12 tasks) while using <= 2% of the same P3 training data. Further improvements are reported for few-shot fine-tuning.

**Summary Of The Review:**

I think this is a reasonable empirical paper, though I have concerns about how well results would generalize outside of P3-style prompts (which tend to be shorter than other instruction tuning type datasets). This combined with the extreme simplicity of the method undercuts some of my enthusiasm for the acceptance, though I think the empirical work is generally solid.

---

> ### Author Response · Authors · 2022-11-17
> **Response to Reviewer zbKf**
>
> Thank you for noting that our method is simple, effective, and an important direction for research! We plan to share our code and data soon.
>
> *Approach Simplicity + P3 prompts biasing results*: We note that we experimented on three datasets **not** in the T0 evaluation suite (DROP, CaseHold, QasperEvidence), for which we wrote templates ourselves, reducing the potential issue of biases from similar prompts across P3. However, we agree it is important to examine the role of prompting/instructions in the retrieval method, and have added results involving SuperNatural Instructions (formally called Natural Instructions V2) targeting this. We encode instances with and without instructions respectively for retrieval, but *train on instances with instructions included in order to precisely test the effect of instructions on retrieval only*. We find that retrieving examples without instructions performs slightly better than with instructions, although both models perform similarly overall and outperform a random sample baseline. This suggests that it is the sample information, rather than the prompt, that provides useful information for cross-task retrieval. We also find that DEFT-trained models are on average able to perform similarly to a T5 model trained on all available data (100 instances per training task in the English split of Super-Natural Instructions). We believe this shows that our approach works well beyond P3.
>
> *Negative Interference*: We add a new baseline, Rand-bal, which uniformly samples from all tasks in P3, and underperforms DEFT (see table 1). The fact that DEFT outperforms Rand-bal, Rand, T0 baselines, which are identical models simply trained on different data, suggests that it is the DEFT data selection that allows better generalization to the target task and that the additional data present in T0 training makes it more difficult for the model to generalize well.
>
> *Model naming conventions*: In regards to the naming confusion, all our models are based on T5 v1.1 + LM adapt models, so 'DEFT-t5-base' is the same as 'DEFT-base', which uses t5-base v1.1 (roughly 250 million parameters in size). DEFT-XL uses t5-xl v1.1 (roughly 3 billion parameters), as you noted. We have cleaned this naming confusion up in the updated paper, and standardized naming to always call DEFT models ‘DEFT-<size>’.
>
> Edit: we have edited this comment to make the links to the points raised by the reviewer clearer.

---

### Author Response · Authors · 2022-11-17
**Paper update based on Reviewer feedback**

We thank all the reviewers for their insightful and useful comments, which have greatly helped improve the paper. We are happy that reviewers were overall positive about our work and noted that the results are strong, interesting, and well-performed, the idea is simple yet effective, and that the paper is clear and well-written. We have uploaded a new version that makes a number of changes, which we list below. We also note where edits were made based on reviewer feedback, or generally done to improve the paper:
- [Reviewer g3S4] *ReCross comparison* - We have made more reference to ReCross, particularly in the introduction (section 2), results (table 1), and discussion (section 4.2). We note that our paper focuses on **pruning multitask data to only relevant subsets for training**, whereas ReCross focuses on **augmenting massively multi-task models via continued training**. We believe this offers a sufficiently novel perspective and utilization of cross-task retrieval, placing it in an entirely different context to ReCross. For more details, please see our response to Reviewer g3S4. **We also note that according to ICLR guidelines, ReCross is considered concurrent work**. Nonetheless, We cite ReCross and add comparisons in our work.
- [Reviewer g3S4] *ReCross results* - We also have added ReCross results in table 1 and find that **ReCross significantly underperforms DEFT** on 4/6 tasks, with DEFT models improving by an average of 20% over ReCross.
- [Reviwers g3S4, zbKf] *Experiments beyond P3* - We have added expanded results involving SuperNatural Instructions (previously called Natural Instructions V2) to explore the effect of instructions/prompts in retrieval. We discuss the results in section 5.1 (‘Are prompts useful for retrieval?’), with detailed results in Appendix D.
- [Reviewer zbKf] *Naming conventions* - We have standardized naming to always call DEFT models ‘DEFT-<size>’. All DEFT models are trained using the named size T5 v1.1 + LM Adapt as the starting point.
- [Reviewer SJym] *Data sampling* - We have added a balanced random baseline for XL size models which uniformly samples from tasks for random selection in table 1.
- [Reviewer SJym] We have added examples of what gets retrieved in Appendix E.
- [Reviewer fvhK] *BM25 performance* - We have added results from running a BM25 baseline in table 1 and found that it does similarly to random data selection, suggesting that a neural retrieval approach is required to find good cross-task neighbors.
- [General] We have expanded the case study in section 5.2 to show the effect of increasing retrieved and labeled data for QasperEvidence.
- [General] We have added the results for the DEFT-few models before parameter-efficient IA3 tuning in Appendix B.
- [General] We have added a comparison between using base and XL-size models for indexing and retrieval in Appendix C and reference this in the main text (section 5.1, ‘What model should you use for retrieval?’).
- [General] We have expanded the datasets tested to include RTE and ANLI R3 (tables 1,2, and figure 2).
- [General] We have replaced the t5-base results in table 1 with results using the original T0 training task pool, making the experimental setting identical to the XL size setting.
- [General] Minor edits to accommodate new results while staying within page limit.

We discuss these additional results and edits where relevant in our responses to each reviewer. We hope that these additions address the concerns and questions raised by each reviewer, and encourage reviewers to respond if they have further queries. We also plan to release our code after the review period in order to aid reproducibility.

edit: we have edited this comment to make it clearer what changes were made based on reviews.

---

### Decision · Program_Chairs · 2023-01-20

**Decision:**

Reject

**Justification For Why Not Higher Score:**

I have concerns regarding the experimental setup of the paper, and the fact that some claims made in the paper might be due to the imbalance of the used dataset (P3), instead of negative interference between tasks. I believe that some additional experiments done during the discussion go in that direction, and thus, a more thorough empirical evaluation is needed before acceptance.

**Justification For Why Not Lower Score:**

None

**Metareview: Summary, Strengths And Weaknesses:**

This paper introduces a new method to improve zero/few-shot generalisation of large language models. Starting from a small set of unlabelled examples, the method retrieves similar labelled examples from a large collections of diverse tasks, which are then used to fine-tune the language model. The authors show that this technique leads to competitive results, compared to fine-tuning the model on the full set of examples, and suggest that there is a negative interference between tasks.

While the reviewers found the results encouraging, and liked the general direction of data curation, they also raised concerns which I tend to share. More specifically, reviewers are concerned by the baselines, which are mostly multi-task, while the proposed method trains a different model for each task. Second, the observed effect might be due to the unbalanced nature of the P3 dataset: when sampling uniformly over tasks (instead of uniformly over examples), the “rand” baseline improved a lot, from 45.9 to 50.4. This lead to the question of how well this baseline would do by increasing the number of examples (instead of limiting to the same number as the proposed method). Similarly, on Super-Natural Instructions, the method does not perform as well as the baseline. For these reasons, I believe that more empirical evidence is necessary to demonstrate the effectiveness of the method, and recommend to reject the paper.